# The globus pallidus orchestrates abnormal network dynamics in a model of Parkinsonism

Brice de la Crompe [1,2], Asier Aristieta[1,2], Arthur Leblois[1,2], Salma Elsherbiny[1,2], Thomas Boraud [1,2] & Nicolas P. Mallet[1,2 ✉]

The dynamical properties of cortico-basal ganglia (CBG) circuits are dramatically altered following the loss of dopamine in Parkinson's disease (PD). The neural circuit dysfunctions associated with PD include spike-rate alteration concomitant with excessive oscillatory spike-synchronization in the beta frequency range (12–30 Hz). Which neuronal circuits orchestrate and propagate these abnormal neural dynamics in CBG remains unknown. In this work, we combine in vivo electrophysiological recordings with advanced optogenetic manipulations in normal and 6-OHDA rats to shed light on the mechanistic principle underlying circuit dysfunction in PD. Our results show that abnormal neural dynamics present in a rat model of PD do not rely on cortical or subthalamic nucleus activity but critically dependent on globus pallidus (GP) integrity. Our findings highlight the pivotal role played by the GP which operates as a hub nucleus capable of orchestrating firing rate and synchronization changes across CBG circuits both in normal and pathological conditions.

[1] Université de Bordeaux, Institut des Maladies Neurodégénératives, 33076 Bordeaux, France. [2] CNRS UMR 5293, Institut des Maladies Neurodégénératives, 33076 Bordeaux, France. ✉email: nicolas.mallet@u-bordeaux.fr

Normal brain functions rely on the overall modulation of cell firing activity and the precise control over the spatiotemporal firing pattern, including (possibly synchronized) oscillatory activity among neuronal network and brain areas[1,2]. Disruption in the dynamical properties orchestrating local firing rates and global network oscillations changes are observed in many neurological disorders[3]. This is particularly well-illustrated in the basal ganglia (BG) where the loss of dopamine in Parkinson's disease (PD) is associated with persistent alterations in both the firing rate[4,5] and oscillatory synchronization among and between BG nuclei[6–8]. Defining how these changes interact and are orchestrated is a key aspect to better understand the circuit mechanism underlying their generation.

On one hand, spike-rate changes are the backbone of the classic model of BG dysfunction[9]. In this scheme, dopaminergic loss in the striatum triggers an imbalance in the firing activity of striatal neurons involved in BG direct and indirect pathways, resulting in a cascade of firing rate changes along these circuits which ultimately leads to an overinhibition of the motor system[4,5]. The direct/indirect striatal imbalance in firing activity has been confirmed experimentally[10,11], and the effect of firing rate alterations on the motor deficits in PD is highlighted by optogenetic studies showing that driving hyperactivity in the indirect pathway striatal neurons generates a parkinsonian-like state in rodent[12]. In addition, spike-rate features can be predicative of parkinsonism[13,14].

On the other hand, changes in BG oscillatory activity are also recognized as a critical functional change associated with PD[7]. It materializes as an excessive expression of synchronized oscillatory activity across BG nuclei in the beta ($\beta$) frequency range (12–30 Hz) in humans or experimental parkinsonism[15,16]. These synchronized oscillations were first associated with tremors[6] but seem to be better correlated with akinesia/rigidity[13,17]. Unlike spike-rate changes, their generation mechanisms remain unknown. From a theoretical perspective, any neuronal networks incorporating negative feedback loops with delays can generate oscillatory activity patterns[18]. The multiple parallel and recurrent inhibitory feedback loops in the BG network include many circuits that, in theory, could generate PD-related $\beta$-synchronization[19–21]. The first BG pattern generator system identified was the reciprocally connected subthalamic nucleus (STN) and globus pallidus (GP) circuit[22,23]. The organization of GP into prototypic and arkypallidal neurons might also constitute a key neuronal substrate to propagate these $\beta$-oscillations in BG loop[24,25]. However, alternative circuit generators have also been proposed in the cortex[26], the striatum[27], and other BG networks[19,28,29]. In particular, the cortex is a strong rhythm generator[30] that presents abnormal $\beta$-synchronization during PD[31]. As such, one attractive hypothesis is that the parkinsonian $\beta$-oscillations might be generated in the cortex and abnormally maintained/amplified by GP–STN network[26,32] or the hyperdirect pathway[21,28]. However, the specific contribution of these different circuit components has never been tested with population-specific and millisecond time-scale control of neuronal networks. In this study, we combined in vivo electrophysiological recordings in normal and parkinsonian rats with optogenetic manipulations (i.e., opto-excitation and opto-inhibition) to dissect the specific functional contribution of key cortico-basal ganglia (CBG) components (i.e., the motor cortex, the STN, and the GP) and understand how the generation/propagation of abnormal firing rate and synchronized oscillatory activity in PD is orchestrated.

## Results

**mCx is not necessary for abnormal $\beta$-dynamics in 6-hydroxydopamine (6-OHDA) rats**. To test the contribution of motor cortex (mCx) in generating abnormal firing rate and synchronization changes in BG, we microinjected an AAV5-CaMKII-ArchT3 virus in mCx areas of 6-OHDA hemi-lesioned rats (Fig. 1a). Opto-inhibition of cortical neurons was confirmed using custom-made opto-electrodes (see 'Methods') and extracellular unit recordings (Fig. 1b). Cortical neurons were all classified as putative pyramidal cells based on action potential duration (Fig. 1c). Most cortical neurons responded to light stimulation with a significant reduction of firing (Fig. 1d). Quantitative analysis of the light-induced inhibition both at the individual (Fig. 1e, Supplementary Table 1) and population level (Fig. 1f) show that the net effect was an overall strong reduction of mCx neuronal firing rate. We next characterized the impact of mCx opto-inhibition on STN firing and the level of $\beta$-synchronization in CBG circuits (Fig. 1g). These experiments were conducted in an activated cortical state that favors the concomitant expression of both $\beta$-oscillations and STN firing hyperactivity[24,33]. We first showed that most STN neurons decrease their activity during mCx opto-inactivation (Fig. 1h) but the inhibitory effect was moderate (Fig. 1i, j, Supplementary Table 1). Interestingly, STN average firing during mCx opto-inhibition in 6-OHDA rats was still faster than the one recorded in control dopamine-intact animals (Fig. 1i, Supplementary Table 1), suggesting that STN firing hyperactivity is not primarily driven by cortical inputs. We then investigated how $\beta$-oscillations' expression was affected by mCx opto-inhibition. Looking first at $\beta$-oscillations' expression in mCx electrocorticogram (ECoG) power spectrums, we found no effect of mCx opto-inhibition (Fig. 1k). The first spectral analysis was limited to the best $\beta$-oscillations recording epochs (as determined by power, see 'Methods') obtained in each rat. Calculation of the 12–30 Hz $\beta$-band areas under the curve (AUC) (Fig. 1l, Supplementary Table 1) and measure of the $\beta$ burst dynamics (Supplementary Fig. 1a–c) reveal no significant effect of mCx opto-inhibition. The power of $\beta$-oscillations fluctuates in vivo between high and low $\beta$ epoch. To reveal if mCx opto-inhibition had an effect on these dynamical properties of $\beta$-oscillations, we quantified the laser effect on each light stimulation by comparing the $\beta$-AUC powers ON versus OFF (Fig. 1m). To control for the dynamical nature of $\beta$-oscillations we also compared the difference between the 2 s $\beta$-AUC epoch directly preceding the OFF epochs (i.e., the 'Pre' epoch) and the $\beta$-AUC power of the OFF epoch. Interestingly, even when observing individual $\beta$ epochs of various power, we did not see any significant effect of mCx inhibition on the $\beta$-AUC$_{(OFF–ON)}$ and $\beta$-AUC$_{(Pre-OFF)}$ distribution function (Fig. 1n, Supplementary Table 1). Finally, we then investigated if mCx opto-inhibition affected the spike-timing properties of STN neurons during $\beta$-oscillations and found that the mean $\beta$-phase locking of STN neurons (Fig. 1o, Supplementary Fig. 1d and Table 1) and the vector lengths (Supplementary Fig. 1e) were not significantly impaired by mCx silencing. Although, the $\beta$-band coherence between STN unit and mCx ECoG was slightly decreased during mCx inhibition (Supplementary Fig. 1f, g), this decrease could be explained by the sensitivity of the spike-field coherence measures to firing rate level[34], as verified by randomly removing action potential in STN spike trains of the same OFF epoch dataset (Supplementary Fig. 1h–j). The contribution of other cortical areas (e.g., somatosensory cortex) in $\beta$-oscillations' generation was also excluded by performing large-scale cortical ablation in 6-OHDA rats (Supplementary Fig. 2a, b). Altogether, these results rule out the contribution of mCx as a major source for the generation and the transmission of $\beta$-oscillations in the 6-OHDA rat model of PD.

**STN is not necessary for abnormal $\beta$-oscillations expression**. STN neurons are highly synchronized at $\beta$-frequencies and this

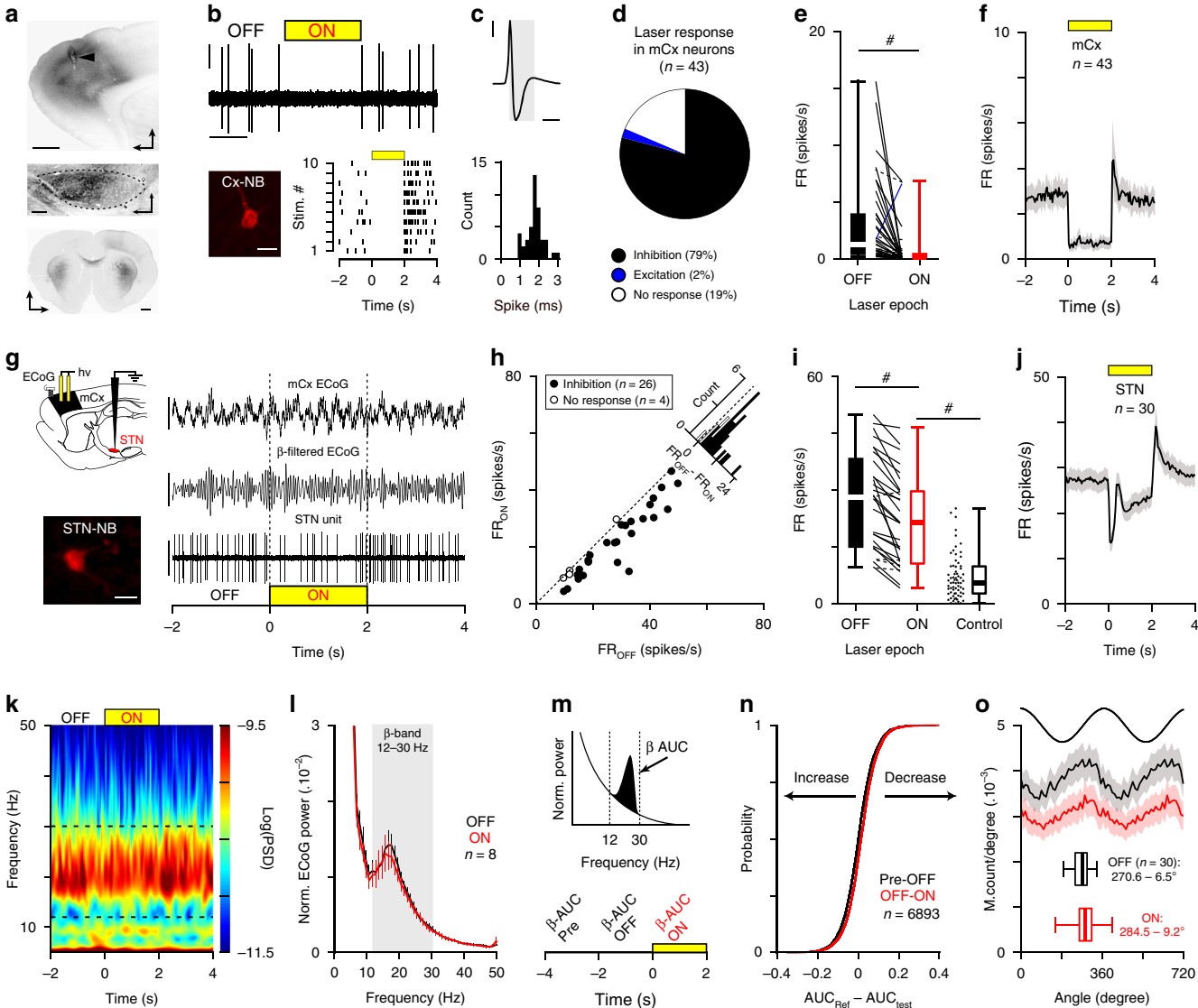

**Fig. 1 Motor cortex activity is not necessary for abnormal network dynamics in cortico-BG loop. a** Epifluorescent images showing eArchT3-EYFP labeling in mCx (top, scale: 1 mm), STN (middle, scale: 200 μm), and striatum (bottom, scale: 1 mm) of 6-OHDA rats. Arrowhead indicates the optic fiber track in mCx. **b** Opto-inhibition example of a pyramidal neuron (top, scales: 0.5 mV, 1 s) labeled with neurobiotin (bottom left, scale bar: 20 μm) and corresponding raster plot (bottom right). **c** Example of action potentials waveform average (top, scales: 0.5 mV, 1 ms, same neuron as in **b**) and spike duration distribution for mCx recorded neurons (bottom, bin: 0.2 ms). **d** Proportion of mCx neurons inhibited (black), excited (blue), or not modulated (white) by laser stimulation. **e** Box-and-whisker plots comparing mCx firing rate during OFF/ON laser epochs; firing rate (FR). **f** Population PSTH in mCx during laser stimulation (bin: 50 ms). **g** Experimental paradigm (top left) and representative example of ECoG (top right, scales: 100 μV), β-filtered ECoG signal (12–30 Hz, middle right, scale: 50 μV), and one STN unit (bottom right, scale: 2 mV) during mCx opto-inhibition. Juxtacellular neurobiotin labeling confirmed STN recordings (bottom left, scale: 20 μm). **h** Scatter plots of STN neurons firing rate during ON versus OFF laser stimulation. **i** Box-and-whisker plots of STN firing during OFF/ON laser epochs in PD rats, and in control animals. **j** Population PSTH of STN firing during mCx opto-inhibition (bin: 50 ms). **k** Peri-event spectrogram of mCx ECoG recording during mCx opto-inhibition. **l** Mean normalized power spectrum of mCx ECoG during OFF versus ON laser epochs (analysis on the best β epochs for each rat). **m** Schematic of the β-AUCs calculation (top) that we performed at three consecutive time periods: 'Pre', 'OFF', and 'ON' laser stimulation (bottom) for every single opto-stimulation epoch. **n** Cumulative distribution function of the β-AUCs differences calculated between AUC$_{(Pre-OFF)}$ and AUC$_{(OFF-ON)}$ for every ECoG β epochs. **o** Mean phase histograms of STN neurons during β-oscillation for the OFF and ON laser epochs. Group data represents mean ± SEM, box-and-whisker plots indicate median, first, third quartile, min, and max values. See also Supplementary Table 1 for statistical information.

activity could be important for orchestrating abnormal β-rhythm[24,25,35]. We tested the STN contribution to β-oscillations' generation using an opto-inhibitory approach (Fig. 2a) based on two different viral constructs: either an AAV5-CaMKII-eArchT3 virus as performed in a seminal study[36], or an AAV5-hSyn-eArch3. Qualitative (Fig. 2b, c) and quantitative (Supplementary Fig. 3a, b) histological control of EYFP expression confirms the transduction of STN neurons. The efficacy of STN opto-

inhibition was verified for each animal using opto-electrode mapping and the animals that did not reach satisfactory level of STN inhibition (i.e., >60%) were excluded from further analysis (see 'Methods'). For both sets of experiments, STN neuronal firing rate was strongly reduced by light stimulation (Fig. 2d, e and Supplementary Table 2). We next tested the impact of STN opto-inhibition on the expression of β-oscillations (best β-power recordings for each rat). We found that the β power (Fig. 2f and

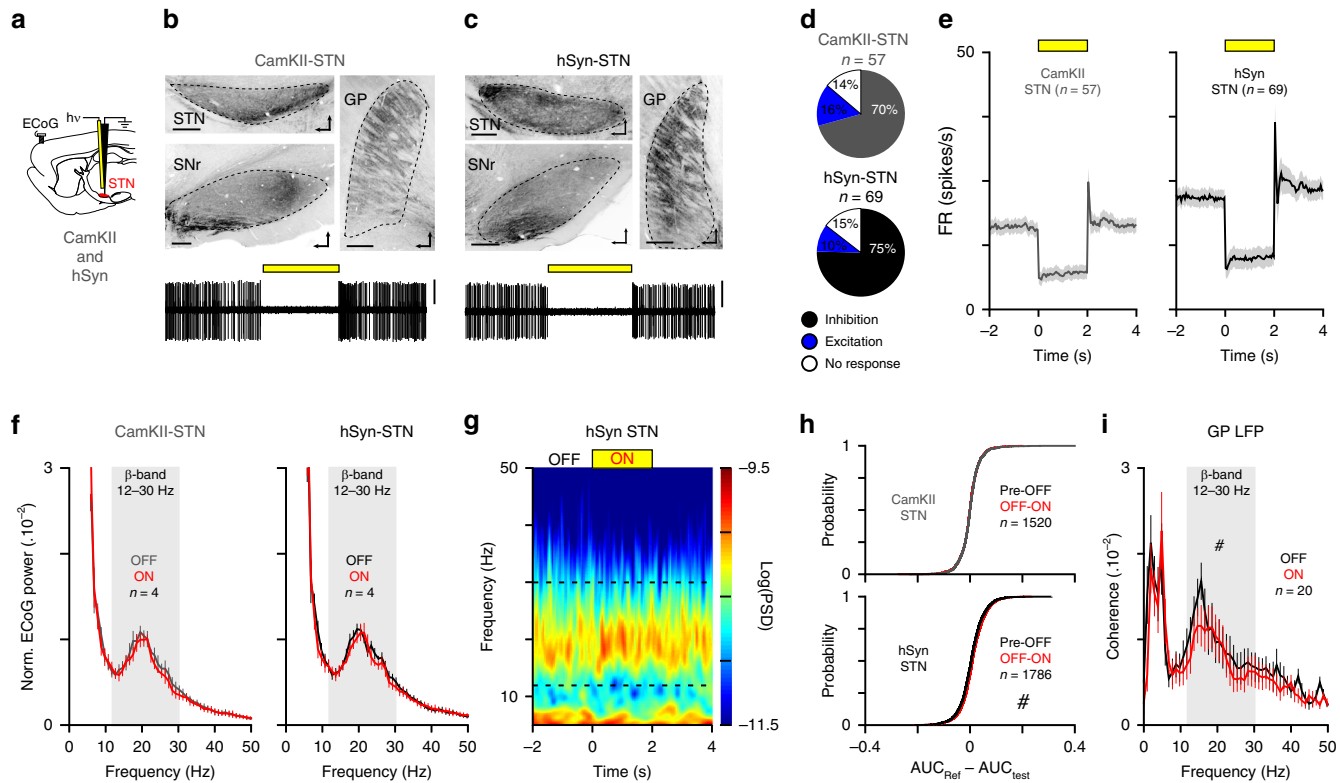

**Fig. 2 STN activity is not necessary for abnormal β-synchronization in cortico-BG loop. a** Schematic of STN opto-inhibition using CamKII or a ubiquitous and neuron-specific promotor: the human synapsin 1 (hSyn) dependent virus in 6-OHDA rats. **b, c** Sagittal epifluorescence images (top) showing the EYFP expression in STN and the appropriate axonal labeling in GP and substantia nigra pars reticulata (SNr). Bottom traces show examples of STN neuron opto-inhibition in CamKII-STN (**b**) or hSyn-STN (**c**) conditions (slice orientation: dorso-rostral, scale: 400 μm, scales: 0.5 mV, 6 s recording). **d** Proportion of STN neurons inhibited (black), excited (blue), or non-modulated (white) by the laser stimulation in CamKII-STN and hSyn-STN. **e** Population PSTH of STN firing during opto-inhibition in CamKII-STN (bin: 50 ms) and hSyn-STN groups (bin: 50 ms). **f** Mean normalized power spectrum of mCx ECoG (analysis on the best β epochs recordings) during OFF and ON laser stimulations in CamKII-STN (left) and hSyn-STN (right). **g** Example of peri-event spectrogram of mCx ECoG recording during STN opto-inhibition in hSyn-STN rat. **h** Cumulative distribution function of the β-AUCs differences calculated for AUC$_{(Pre-OFF)}$ versus AUC$_{(OFF-ON)}$ for all ECoG β epochs in CamKII-STN (top) and hSyn-STN (bottom) conditions. **i** Mean coherence for ECoG versus GP LFP at β-frequency during OFF and ON laser stimulation in hSyn-STN rats. Group data are mean ± SEM. See also Supplementary Table 2 for statistical information.

Supplementary Table 2) and the β burst duration (Supplementary Fig. 3c) were not affected by STN opto-inhibition whereas the β burst count was slightly decreased (Supplementary Fig. 3d). To analyze if STN opto-inhibition had an effect on β dynamics, we next compared all the individual β epochs in control versus laser conditions and found either no laser effect in CaMKII-eArchT3 or a small reduction in the hSyn-eArchT3 group (Fig. 2h and Supplementary Table 2). We also performed GP LFPs recordings in hSyn-eArchT3 rats to identify if β-synchronization was affected in BG circuits. We found that although the coherence between GP LFPs and mCx ECoG was significantly reduced (Fig. 2i and Supplementary Table 2) it was nevertheless not suppressed by STN opto-inhibition. The lack of strong effects on β dynamics consequent to STN opto-inhibition was further confirmed by the results of STN ablation. Indeed, we found that electrolytic lesion of the STN did not affect the level of β-oscillations expression in lesioned rats (Supplementary Fig. 4a, b). In conclusion, our data reveal that, against the current assumption, STN activity in 6-OHDA rats is not involved in the generation of abnormal β-oscillations and, at best, only play a supportive role.

## Opto-patterning of STN neurons at β-frequency in normal rats. The inability to fully suppress STN activity during our opto-inhibition protocol could, in theory, explain the absence of detectable effect on β-oscillation expression. To further test if

STN neurons are important for β-generation, we artificially introduced β-oscillations directly at the level of STN in normal animals and dissected the neuronal substrates of these synthetic β-oscillations in comparison with the parkinsonian β. Reverse engineering of β-oscillations was achieved by stimulating rhythmically the activity of STN neurons at β-frequency (20 Hz) using an optical excitatory (AAV-CaMKII-ChR2, Fig. 3a, b) or an inhibitory approach (AAV-CaMKII-eArchT3, Supplementary Fig. 5a, b). These excitatory/ChR2 and inhibitory/ArchT β-patterning strategies were also used to dissect the functional contribution of specific excitatory versus inhibitory inputs to STN neurons. We first validated these tools using opto-electrode mapping in STN. As expected, opto-patterning STN neurons at β-frequency artificially synchronized STN neurons activity at 20 Hz (Fig. 3e, Supplementary Fig. 5f) and this activity could propagate to generate abnormally high level of cortical β-oscillations (Fig. 3c, Supplementary Fig. 5d and Table 3) and cortico-STN coherence (Supplementary Fig. 6a, b). Interestingly, these STN opto-patterning protocols were more efficient at generating abnormal β-synchronizations (as measured in the ECoG) when the brain state of the animal was activated as opposed to slow-wave oscillations. This property parallels the brain-state dependency of parkinsonian β-oscillations expression already described in anaesthetized[24] and awake[33] 6-OHDA lesioned rats. As such, these opto-patterning experiments were preferentially performed in an activated brain state. Combining STN-ChR2 β-patterning

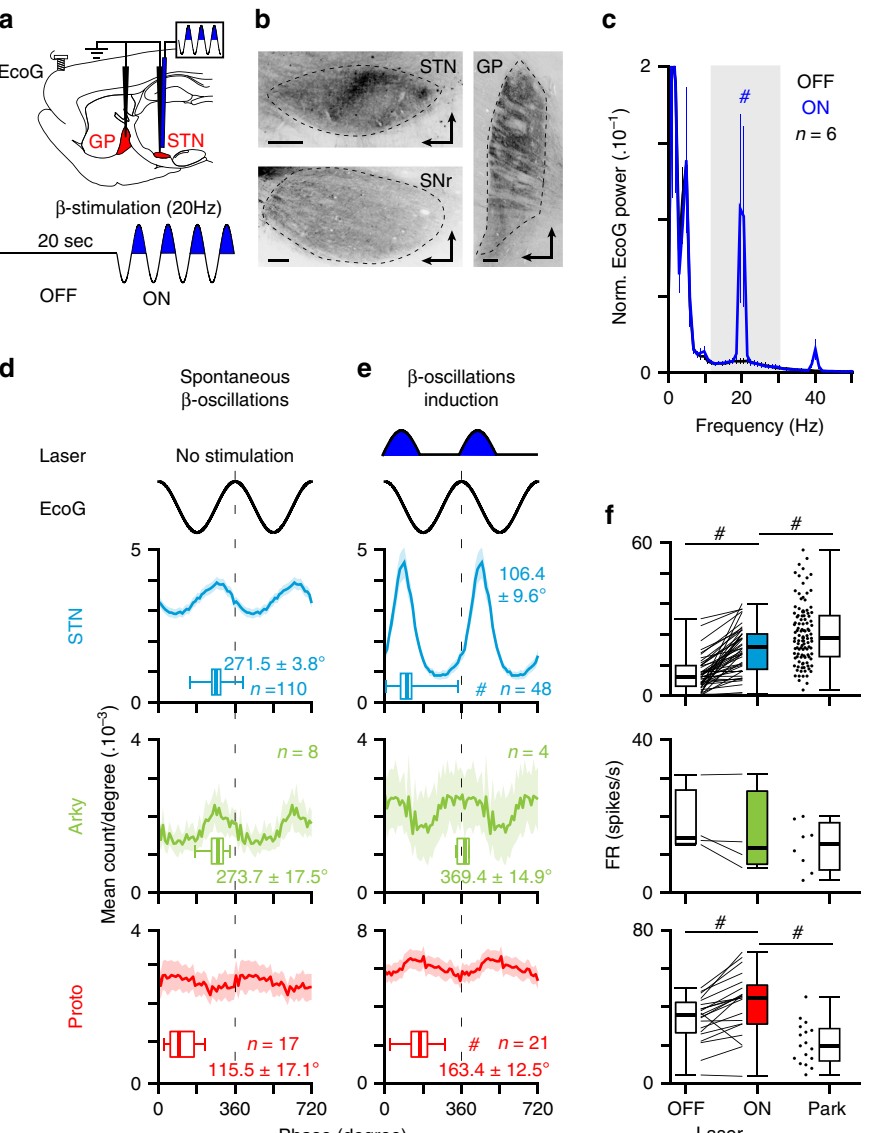

**Fig. 3 Excitatory optogenetic patterning of STN neurons at $\beta$-frequency does not replicate the functional properties of parkinsonian $\beta$-oscillations.**
**a** Schematic of the experiment in ChR2-expressing STN neurons (top) and laser stimulation protocol (bottom) used to mimic abnormal $\beta$-oscillations in normal rats. **b** Sagittal epifluorescence images showing the ChR2-EYFP labeling at the level of the STN (top left) and their axons in SNr (bottom left) and GP (right) (scale: 200 μm). Slices orientation: dorso-rostral. **c** Mean power spectrum of mCx ECoG during OFF and ON laser stimulation in normal rat. **d** Mean phase histograms of STN (top), arkypallidal (middle), and prototypic (bottom) neurons during abnormal $\beta$-oscillations recorded in 6-OHDA-lesioned rats. **e** Mean phase histograms of STN (top), arkypallidal (middle), and prototypic (bottom) neurons during synthetic $\beta$-oscillations evoked via opto-patterning of STN neurons using ChR2 in control rats. **f** Comparison of the change in firing rate induced by synthetic $\beta$ as compared with abnormal parkinsonian $\beta$ in STN, arkypallidal, and prototypic neurons. Group data represents mean ± SEM, box-and-whisker plots indicate median, first and third quartile, min, and max values. See also Supplementary Table 3 for statistical information.

with unit activity recordings of the two main population of GP neurons (i.e., the prototypic and arkypallidal cells) also reveal that synthetic $\beta$-oscillations could propagate to GP neurons and cause a significant increase in the temporal coupling (measured through coherence analysis) between cortex and the two populations of GP neurons (Supplementary Fig. 6a, b). We next assessed if our synthetic $\beta$-synchronization shared the same or as many as possible functional properties as parkinsonian $\beta$-oscillations. In particular, we focused our analysis on the known STN/GP neurons spike-timing properties during these oscillations (that is the antiphase relationship between STN/prototypic and prototypic/arkypallidal neurons, as well as the in-phase relationship between STN/arkypallidal neurons[24,25]) and the associated firing rate changes (that is the hyperactivity of STN neurons versus the

decrease firing of prototypic neurons). Importantly, these fine STN/GP spike-timing features during PD $\beta$-oscillations were confirmed in our dataset (Fig. 3d). On the contrary, artificial ChR2-induced $\beta$-oscillations did not reproduce any of the correct STN/GP phase-locking preferences (Fig. 3e and Supplementary Table 3) while ArchT-induced $\beta$-oscillations reproduced the correct phase-locking value of STN neurons but not the one of GP (Supplementary Fig. 5f). Importantly, both these artificial $\beta$-oscillations did not replicate the stereotypical antiphase firing relationship between STN and prototypic GP neurons (Fig. 3d and Supplementary Fig. 5f, g). Instead, we found that the phase lags between STN and prototypic neurons (~57° for ChR2, Fig. 3e versus ~42° for ArchT, Supplementary Fig. 5g) represent a time delay of 7.9 and 5.8 ms, respectively, compatible with a

monosynaptic excitatory drive from STN inputs to prototypic neurons[37]. Interestingly though, in both cases, we could reproduce (albeit with different phase-locking values toward the cortical β-cycle reference) the antiphase relationship between prototypic and arkypallidal neurons (Fig. 3d, e and Supplementary Fig. 5g) suggesting that this opposition of phase is likely driven by prototypic neurons collateral inhibition onto arkypallidal cells. These firing relationships between STN/GP neurons were also confirmed using longer (2 s) continuous light stimulation in the ChR2 dataset (Supplementary Fig. 6c–e). Finally, considering firing rate changes induced by the ChR2 or the ArchT β-patterning, we found that, in both cases, artificial β-oscillations could not faithfully reproduce the rate changes associated with parkinsonian β-oscillations (i.e., STN hyperactivity and prototypic neurons hypoactivity[24,25]). Indeed, here ChR2 opto-patterning of STN neurons at β-frequency reproduced STN increase firing (Fig. 3f and Supplementary Table 3) but also induced hyperactivity of prototypic cells (Fig. 3f and Supplementary Table 3), whereas ArchT reduced cell firing in both STN and GP neurons (Supplementary Fig. 5h). Taken altogether, these results obtain in lesioned and control animals invalidate the causal importance of STN neurons in driving abnormal BG network dynamics in PD rodents from specific excitatory or inhibitory inputs.

**GP opto-inhibition suppresses abnormal β-synchronization**. The dichotomous organization of the GP into prototypic and arkypallidal neurons has been proposed as a key neuronal substrate in PD to maintain and propagate abnormal β-synchronization to the whole BG circuit[25]. To test if GP activity was important for β-oscillations expression, we applied a global opto-inhibition approach using an AAV5-hSyn-eArch3-EYFP virus. Histological verification confirms the transduction of GP neurons (Fig. 4a). We next verified the laser inhibitory effect and spread in GP using an opto-electrode made of two optic fibers (placed at 1, or 2 mm from the recording tip). Prototypic and arkypallidal neurons were identified through their distinctive electrophysiological signature (Fig. 4b)[25,38]. Extracellular in vivo recordings showed that most putative prototypic neurons were significantly inhibited by light stimulation placed at 1 mm (Fig. 4c–e and Supplementary Table 4) or 2 mm above the recording electrode (Supplementary Fig. 7a, c). Putative arkypallidal neurons were also strongly inhibited by light stimulation both from the 1 mm (Fig. 4d, f and Supplementary Table 4) or 2 mm spaced fibers (Supplementary Fig. 7b, d). Overall, these in vivo GP recordings confirm that our opto-stimulation had a strong and global suppressive effect on both GP neuronal populations. We next assessed the functional impact of GP opto-inhibition onto STN firing activity and mCx ECoG during abnormal β-synchronization (Fig. 4g). We first found that GP opto-inhibition produced a strong excitatory response in most STN neurons (Fig. 4h–j and Supplementary Table 4), highlighting the powerful tonic inhibitory control exerted by prototypic GP neurons onto STN activity. Similarly, β power analysis in mCx ECoG during GP opto-inhibition revealed a strong suppression of β-oscillations expression (Fig. 4k). Both β-AUCs computed on the best β recordings (Fig. 4l and Supplementary Table 4) and on all individual β epochs (Fig. 4m and Supplementary Table 4) were significantly reduced in laser versus control conditions. The β burst dynamics (i.e., duration and counts) were also strongly disrupted (Supplementary Fig. 7i, j). Furthermore, at the population level, the STN β-phase locking histogram was strongly suppressed by GP opto-inhibition (Fig. 4n). At individual neuronal level, all phase-locked STN neurons had a significant decrease in their vector lengths value (Supplementary Fig. 7e, f)

and their strength of entrainment (only 9 out of the 35 phase-locked neurons were still significantly entrained with a Rayleigh test value $p < 0.05$). Also, STN spike-field coherence measures (using mCx ECoG as field) were significantly suppressed by GP opto-inhibition (Supplementary Fig. 7g, h). In conclusion, our results strongly support the idea that GP activity is important and required for the generation of β-synchronization in 6-OHDA rats.

**Change in STN firing do not cause β-rhythm suppression**. How changes in firing rate impact neuronal synchronization is unclear. It is thus possible that the strong increase in STN firing caused by GP disinhibition, impairs the entrainment capacity of STN neurons (i.e., single-cell level effect) or, alternatively, provides a profound change in the excitatory state of the whole BG circuit (i.e., network level effect) that might indirectly contribute to β-synchronizations disruption. To account for both these possibilities, we tested two approaches aimed to either mimic the increase of STN firing rate either at the single-cell level (i.e., juxtacellular approach) or at the network level (i.e., opto-excitation of STN). Juxtacellular stimulation of STN neurons using current injection (Fig. 5a) was sufficient to significantly increase the firing activity of single STN neurons during β-synchronization (Fig. 5b). The intensity of current injection was adapted for each STN neuron to induce a sufficient excitation level (Fig. 5c and Supplementary Table 5). Quantification of STN neurons' frequency modulatory index (MI) showed that the excitatory effect induced by current injection was similar (albeit slightly higher) than the one obtained following GP opto-inhibition (Fig. 5d and Supplementary Table 5). Such depolarization, at the single-cell level, did not affect the capacity of STN neurons to be phase-locked to β-oscillations (Fig. 5e and Supplementary Table 5). Indeed, all excited STN neurons were still entrained by β-oscillations during the ON-juxtacellular epochs with no effect on the mean phase angles (Supplementary Fig. 8a) and a slight diminution of the vector lengths (Supplementary Fig. 8c). Also, the β-band coherence between STN unit and mCx ECoG was significantly increased during juxtacellular stimulation (Supplementary Fig. 8e, g), and this effect was likely due to the increase in the firing rate (see Supplementary Fig. 1i). To test if a global change in BG network dynamical state was responsible for β-synchronization disruption, we performed global STN opto-excitation (Fig. 5f). Qualitative histological observation (Fig. 5g) revealed sufficient STN transduction while our in vivo recordings confirmed the excitatory effect on STN neurons (Fig. 5h). We could thus reproduce using opto-stimulation an overall level of STN excitation similar to the one obtained with GP silencing (Fig. 5i) and the change in MI were not statistically different in both conditions (Fig. 5j and Supplementary Table 5). This strong increase in STN excitatory drive did not impair, nor boost β-oscillations power in mCx (Fig. 5k and Supplementary Table 5). The β-phase locking of STN neurons was slightly shifted to early phases (Fig. 5l and Supplementary Table 5) but most STN neurons were still significantly entrained by β-oscillations ($n = 23/27$, Rayleigh test $p < 0.05$) with a small decrease of both mean phase angles and vector lengths (Supplementary Fig. 8b, d). The β-band coherence between STN unit and mCx ECoG was not affected by light stimulation (Supplementary Fig. 8f, h). These results support the idea that β-oscillations suppression following GP opto-inhibition was not caused by a dynamical change in STN excitatory state but rather due to the loss of GP contribution in orchestrating β-rhythm across CBG circuits.

**Opto-patterning of GP neurons at β-frequency in normal rats**. During natural β-oscillations, the tonic firing activity of prototypic neurons is strongly decreased as compared with control

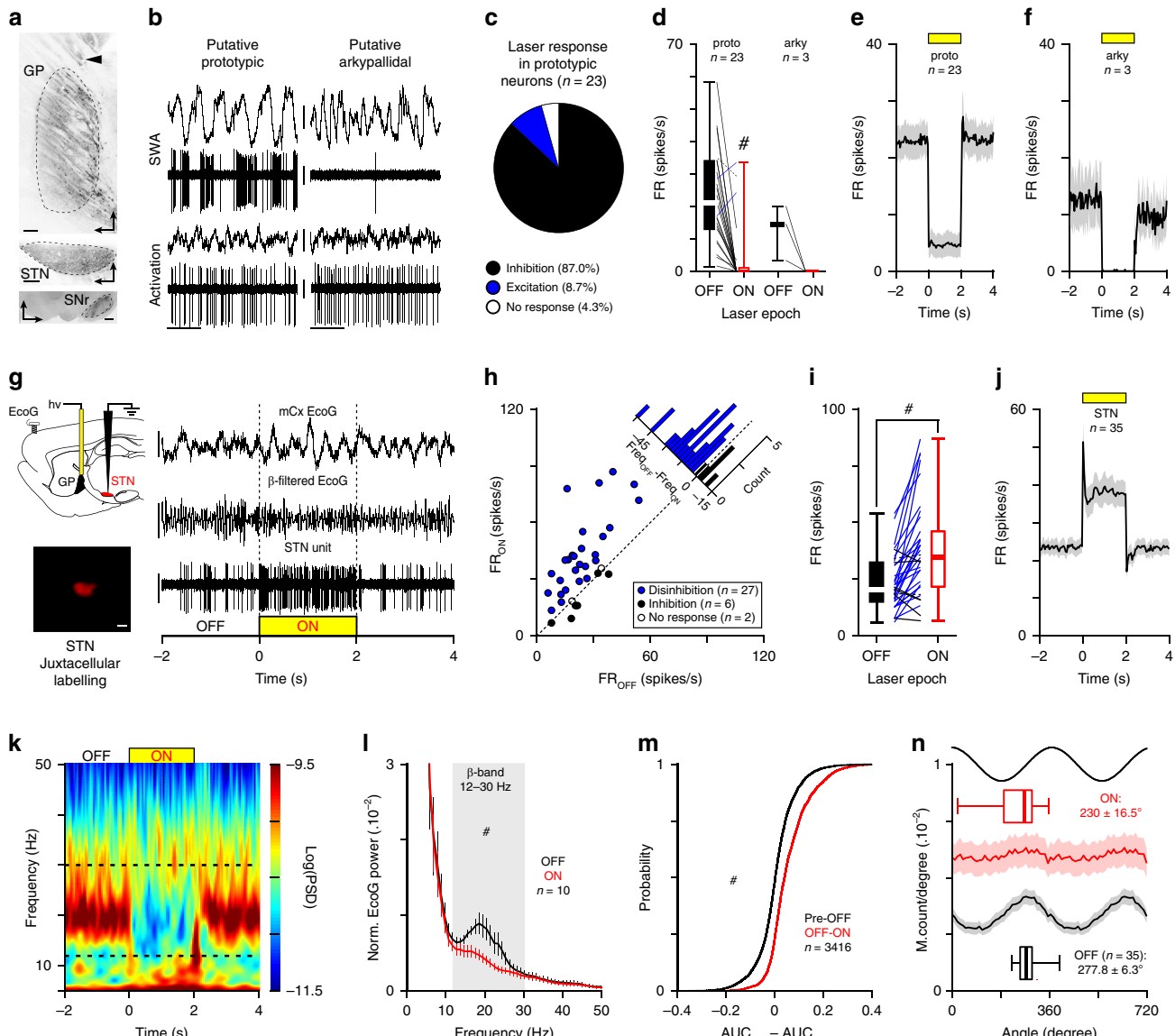

**Fig. 4 Globus pallidus opto-inhibition suppresses parkinsonian abnormal β-synchronization in cortico-BG loop. a** Epifluorescence images showing the eArch3-EYFP labeling in GP of 6-OHDA rats (top, scale: 200 µm, arrowhead indicates the optic fiber track) and dense axonal labeling in STN (middle, scale: 200 µm) and SNr (bottom, scale: 500 µm). **b** Electrophysiological activity of prototypic and arkypallidal neurons during SWA (top) and global activation (bottom). ECoG scale: 0.2 mV; spike unit: 0.5 mV. **c** Proportion of GP neurons inhibited (black), excited (blue), or not modulated (white) by laser stimulation. **d** Effect of opto-inhibition on prototypic and arkypallidal firing. Population PSTH of opto-inhibition on prototypic (**e**) and arkypallidal neurons (**f**) (bin: 50 ms). **g** Experimental paradigm (top left) and representative recordings of ECoG (scales: 250 µV), β-filtered ECoG (12–30 Hz, scale: 250 µV), and one STN unit (scale: 1 mV) during GP opto-inhibition. Example of juxtacellularly labeled STN neuron (bottom left, scale: 20 µm). **h** Scatter plot of GP opto-inhibition on STN firing. **i** STN firing rate changes during OFF versus ON GP opto-inhibition. **j** Mean PSTH of STN during laser stimulations (bin: 50 ms). **k** Peri-event spectrogram of mCx ECoG illustrating the impact of GP opto-inhibition during β-oscillations. **l** Mean normalized power spectrum of mCx ECoG OFF versus ON GP opto-inhibition (analysis on the best β epochs). **m** Cumulative distribution function of the β-AUCs differences calculated for $AUC_{(Pre-OFF)}$ versus $AUC_{(OFF-ON)}$ for all ECoG β epochs during GP opto-inhibition. **n** Mean phase histograms of STN neurons during β-oscillations OFF versus ON laser epochs. Group data represents mean ± SEM, box-and-whisker plots indicate median, first and third quartile, min and max values. See also Supplementary Table 4 for statistical information.

animals[25,39] despite receiving more excitation from a hyperactive STN[33]. In addition, GABAergic D2-MSNs neurons, which increase their activity following DA loss[10], become selectively entrained at β-frequency[40]. These data suggest that inhibitory striatal inputs rather than STN excitation might shape GP abnormal neuronal dynamics (i.e., decrease in firing rate and β-synchronization). We tested this hypothesis using an opto-inhibitory patterning strategy to artificially introduce β-oscillations at the level of GP neurons (Fig. 6a). The injection of

AAV5-hSyn-eArch3 virus was performed in the GP of normal rats and confirmed by histological verification (Fig. 6b). Rhythmic optogenetic inhibition of GP neurons at 20 Hz induced β-oscillations expression in mCx (Fig. 6c and Supplementary Table 6) and increase β frequency coherence of STN/GP neurons toward mCx (Supplementary Fig. 9a, b) which was due to strong β-phase locking of STN/GP neuronal populations (Fig. 6e). As previously, we compared the spike-timing relationship and phase-locking values induced by this synthetic β (Fig. 6e and Supplementary

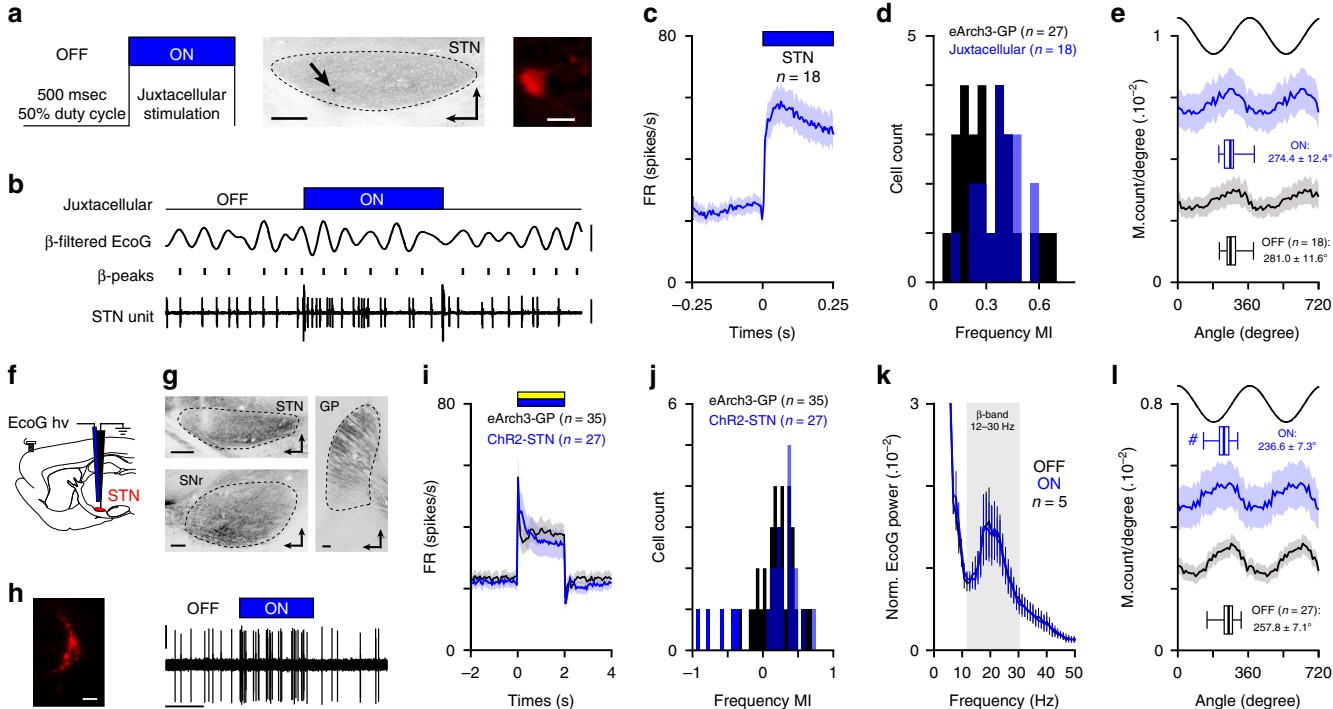

**Fig. 5 Change in STN firing rate does not suppress abnormal β-synchronization. a** Juxtacellular stimulation protocol used to increase the firing rate of single STN neuron as revealed by neurobiotin labeling on sagittal STN epifluorescence image (left image: arrow indicate cell body labeled, scale: 200 μm; right image: neurobiotin-labeled STN neuron, scale 20 μm). **b** Example of STN neuron firing increase during juxtacellular stimulation (scales bars are β-filtered ECoG: 50 μV, spike: 2 mV, time: 750 ms). **c** Population PSTH of all juxtacellularly excited STN neurons recorded during β-oscillations epochs (bin: 5 ms). **d** Comparison of the modulatory index (MI) distribution obtained from juxtacellularly excited STN neurons as compared with disinhibited STN neurons from eArch3-GP opto-inhibition. **e** Mean phase histograms of the juxtacellularly excited STN neurons during β-oscillation during OFF versus ON-juxtacellular stimulation epochs. **f** Schematic illustration of the electrophysiological recordings in STN expressing ChR2-EYFP in 6-OHDA rats. **g** Sagittal epifluorescence images illustrating the ChR2-EYFP signal at the level of STN (top left) and axon terminals in SNr (bottom left) and GP (right). Scale bar represent 200 μm. **h** Example of STN opto-excitation during laser stimulation (right, scales bars are: 0.5 mV, 1 s) and labeled with neurobiotin (left, scale bar is 20 μm). **i** Comparison of population PSTH between all ChR2-excited STN neurons and all STN neurons during eArch3-GP opto-inhibition (bin: 50 ms). **j** Distribution of the MI obtained from ChR2-excited STN neurons and disinhibited STN neurons during eArch3-GP inhibition. **k** Mean normalized power spectrum of mCx ECoG during OFF and ON laser epochs (analysis on the best β epochs for each rat). **l** Mean phase histograms of the STN neurons during OFF and ON laser epochs. Group data represents mean ± SEM, box-and-whisker plots indicate median, first and third quartile, min and max values. See also Supplementary Table 5 for statistical information.

Table 6) with parkinsonian β (Fig. 6d and Supplementary Table 6) for prototypic, STN, and arkypallidal neurons. In contrast to the results obtain with the STN β-patterning, we found here that the phase-locking values of prototypic and STN neurons towards synthetic cortical β-oscillations were similar to the one measured in 6-OHDA rats. Importantly, we could also reproduce the antiphase firing typically present between prototypic and STN neurons in PD condition, suggesting that STN β-synchronization is shaped as a consequence of GP prototypic activity and not the reverse. However, unlike our previous STN opto-patterning experiments, GP opto-inhibition did not reproduce the opposition of phase classically present between arkypallidal and prototypic neurons during β-rhythm. This was likely caused by the fact that our GP inhibitory effect was global and affected both the activity of arkypallidal and prototypic neurons. This finding hints towards the possibility that the inhibitory inputs shaping the parkinsonian β-oscillations preferentially impact onto prototypic neurons (as did STN inputs) and not arkypallidal neurons. In addition, considering firing rates, these β-pattern optogenetic manipulations of GP reproduced the directional firing rate changes classically described in prototypic and STN neurons during parkinsonian β-oscillations, that is: a significant decrease in prototypic activity and a significant increase in STN activity. The inhibitory influence of GP neurons on STN firing was also

evident using longer (2 s) and continuous light stimulation (Supplementary Fig. 9c). Indeed, as in 6-OHDA lesioned rats, the baseline STN firing rate level in control animals was strongly dependent upon GP activity (Supplementary Fig. 9d, e). These experiments also reveal the existence of an inverse correlation between the baseline firing rate of STN neurons (i.e., frequency OFF) and the disinhibition amplitude induced by GP opto-inhibition (Fig. 6g). Interestingly, this correlation was weaker in PD rats as compared with control animals (Fig. 6g and Supplementary Table 6), thus arguing that the control of STN neurons firing rate by GP neurons was partly occluded in 6-OHDA lesioned animals as evinced by the higher baseline firing of STN neurons. Altogether, these results show that the generation of abnormal β-synchronization requires a state of increase incoming inhibition to GP prototypic neurons and that prototypic neuronal activity plays a key role in controlling firing rate level and orchestrates neuronal synchronization in STN neurons in both normal and pathological conditions.

## Discussion

In this work, we used optogenetic stimulation, which provides fast, reversible, and population-selective neural manipulation to shed light on the functional contribution of mCx, STN, and GP to

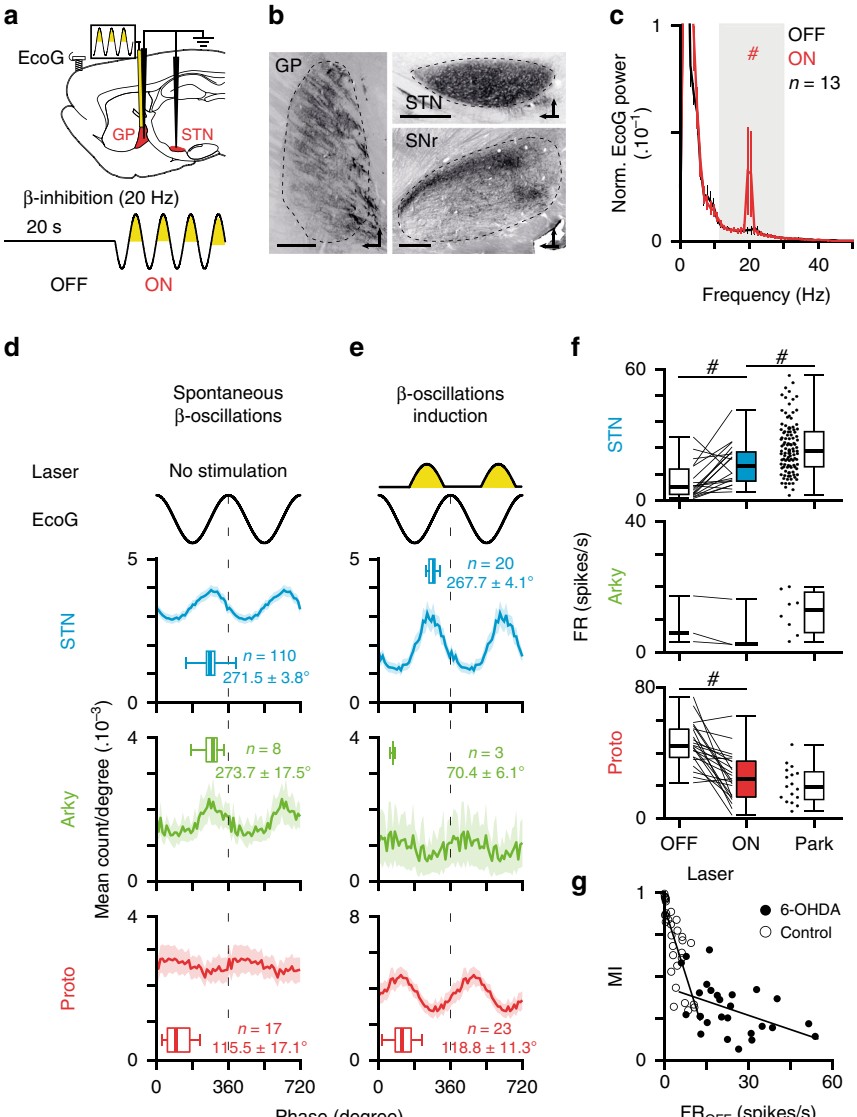

**Fig. 6 Optogenetic patterning of GP neurons at β-frequency replicates the core functional properties of parkinsonian β-oscillations. a** Schematic of the experiments (top) and laser stimulation protocol used (bottom) to reintroduce abnormal β-oscillations in control rats. **b** Sagittal epifluorescence image of a representative eArch3-EYFP expression in GP (right) and target structures in STN (top left) and SNr (bottom left). Scale bars represent 400 μm. **c** Mean normalized power spectrum of mCx ECoG during OFF and ON laser stimulations. **d, e** Mean phase histograms of STN (top), arkypallidal (middle), and prototypic (bottom) neurons during parkinsonian β-oscillations recorded in PD rats (**d**) or during synthetic β-oscillations generated via GP neurons opto-patterning using eArch3 in control rats (**e**). **f** Comparison of the firing rate changes induced by synthetic β as compared with parkinsonian β in STN, arkypallidal, and prototypic neurons. **g** Scatter plots and linear regression of the MI versus frequency OFF calculated for all STN neurons recorded in control animals (open circle) or parkinsonian rats (black circle). Group data represents mean ± SEM, box-and-whisker plots indicate median, first and third quartile, min and max values. See also Supplementary Table 6 for statistical information.

the expression of CBG neural dynamics in control and hemi-parkinsonian rodents.

One influential hypothesis suggests that β-oscillations are generated from cortical sources in normal and pathological conditions, which then propagates to BG circuits (i.e., particularly through the cortico-STN 'hyperdirect' pathway) where they become abnormally amplified in parkinsonism to reach pathophysiological levels[8]. Functional coupling analyses, applied to data recorded from PD patients[41,42] and animal models of parkinsonism[43] have shown that cortical β-oscillations precede in time STN abnormal oscillatory activity, arguing for a cortical drive. However, such phase-differences analyses might not necessarily vouch for causal interactions. Here, we directly tested the cortical β-generation hypothesis using widespread mCx

opto-inhibition in a rat model of PD (Supplementary Fig. 10a). We found that mCx inactivation or large-scale cortical ablation did not alter the level of abnormal β-oscillations expression in CBG circuits. Opto-inhibition of mCx also had a weak effect on STN firing rate hyperactivity which is not entirely surprising. First, STN neurons have autonomous pacemaker firing properties[44], and there is good evidence that the cortico-STN transmission is profoundly reduced in both rodent and monkey PD models[45,46]. The capacity of parkinsonian β-rhythm to synchronize BG circuits in a cortex-independent manner could therefore represent a novel pathological feature of these oscillations. Indeed, if cortical activity can no longer control spike-rate and synchronization changes in BG circuits, this could certainly impair the information processing flow, leading to functional

deficits. Re-establishing cortical control over BG activity might restore some level of functionality in these circuits and could potentially explain the reduction of parkinsonian signs observed following cortical deep brain stimulation[31,36,47]. Altogether, our results argue against a cortical origin of parkinsonian $\beta$-oscillations and rule out a significant contribution of the hyperdirect pathway in driving STN hyperactivity or $\beta$-oscillatory activity. Importantly though, these results do not exclude the possibility that the spontaneous and transient $\beta$-oscillations classically recorded in dopamine-intact conditions during movements are generated through cortical mechanisms[48,49]. Indeed, it remains to be addressed whether parkinsonian $\beta$-oscillations originate from dysregulation in space and time of normal $\beta$-oscillations[50,51] or whether they are generated through entirely different circuit mechanisms.

The STN firing rate increase and synchronized $\beta$-oscillatory activity represent a stereotypical hallmark of parkinsonism[33,35], and these changes could contribute to a stronger functional impact of STN neurons onto CBG activity in the PD state[35]. In addition, the STN is reciprocally connected to GP neurons and this microcircuit forms a negative feedback loop that has been proposed to generate[22], amplify[23], or propagate synchronized oscillatory activity in vivo[24] (Supplementary Fig. 10b). The intrinsic properties of STN neurons could also reinforce abnormal oscillatory activity through post-inhibitory rebound activity[52,53]. However, we show here that STN opto-inhibition, opto-excitation, and lesion had very limited effect on parkinsonian $\beta$-oscillation expression. In addition, reintroducing $\beta$-oscillations directly at the level of STN in normal animals (using $\beta$-frequency opto-patterning) did not mirror the firing rate alterations measured in GP prototypic and arkypallidal neurons in parkinsonian rats[39]. Furthermore, it did not reproduce the correct spike-timing relationships classically observed in the PD condition between STN neurons, prototypic, and arkypallidal neurons[24]. Instead, the phase relationships we obtained when opto-exciting or opto-inhibiting STN inputs are consistent with a preferential functional impact onto prototypic neurons (causing an in-phase relationship with STN), and an opposite response in arkypallidal neurons (causing an antiphase relationship) likely driven by disynaptic inhibition. The excitatory response induced in prototypic neurons upon STN activation has a short synaptic delay (<6 ms) compatible with a direct monosynaptic STN drive[37], whereas arkypallidal inhibition is likely driven by prototypic axon collateral[25]. Such preferential integration of excitatory STN inputs onto prototypic versus arkypallidal neurons could reflect a differential synaptic organization of STN inputs and/or a difference in collaterals inhibitory balance between the two populations of GP neurons. It is tempting to speculate that the differences of spike-timing in STN/GP network during parkinsonian $\beta$ versus artificial $\beta$-oscillations could be due to a reorganization of STN axonal connectivity onto GP neurons induced by the lack of dopamine which can trigger many adaptive changes affecting the functional connectivity within CBG circuits[54,55]. However, the antiphase activity of GP prototypic neurons and STN neurons present after dopamine lesions is also observed in dopamine-intact animals during synchronous spike-and-wave discharges that underlie the absence epileptic seizures[56]. This suggests that the temporal structure of STN/prototypic spike interactions during abnormal synchronized oscillations is imposed by GP prototypic inhibitory activity and not by STN excitatory drive. Altogether, these results cast doubt on the capacity of STN neurons to generate, propagate, or amplify abnormally synchronous oscillatory activity at $\beta$-frequency in CBG circuits in a rodent model of PD.

The functional importance of GP neurons has dramatically changed in the past two decades from being a simple relay nucleus in the indirect pathway[5] to being a central and integrative hub within BG circuits[57,58]. In the PD pathological state, the dichotomous organization of GP neurons has been proposed to play a key role in coordinating and broadcasting pathological network activity[24,25,59] (Supplementary Fig. 10c). Here, we demonstrate for the first time that GP activity is indeed necessary for the expression and propagation of abnormal $\beta$-oscillations in a rodent model of PD. We also show that GP prototypic neurons control baseline STN firing activity both in control and PD conditions. This strong and tonic inhibitory influence likely extends to other nuclei both within and outside BG, especially considering the diverse and widespread projections of GP neurons[25,60]. Interestingly though, the disinhibitory effect induced in STN by GP opto-inhibition was smaller in PD as compared with control animals. These differences could be due to the decrease in prototypic neuronal firing in the PD state that contributes to STN hyperactivity and may thus partially occludes the disinhibition induced by GP opto-inhibition. As such, the synaptic adaptation that has been described for prototypic neurons (i.e., increase strength and number of synapses in STN)[54,61] might thus be a compensatory phenomenon to maintain effective inhibition onto STN neurons. The origin of the prototypic firing rate decrease observed during $\beta$-oscillations is an important question. Given the hyperactivity of STN induced by DA loss and that it preferentially excites prototypic neurons, the most likely cause of prototypic firing reduction is the concurrent increase in inhibition coming from striatal indirect projection neurons[10,11,40]. Striatal indirect neurons become selectively entrained during $\beta$-oscillations[40] with a phase preference that precede STN activity by 45° (which corresponds to ~6 ms assuming a 50 ms $\beta$-cycle period)[24] so striatal $\beta$-inhibition and STN $\beta$-excitation will hit the GP at around the same time. The total number of striatal neurons far exceed those in STN[62] and this is also true for the number of synaptic contacts established in the GP[63,64]. Considering these figures, the synaptic ratio of striatal versus STN inputs is ranging from 30:1 to 50:1, supporting the idea that GP activity might be primarily controlled by striatal inhibition during excessive neuronal synchronization. We confirm here that rhythmic inhibition of GP neurons reproduces the correct spike-timing and firing rate relationship classically measured during parkinsonian $\beta$-oscillations between STN and prototypic neurons. The importance of increased striatal inhibition to induce abnormal oscillatory activity in GP has been suggested by many computational studies[65,66] but how this increase leads to the generation of synchronization at $\beta$-frequency is not currently clear. One possibility is that increased striatal inputs trigger a switch of GABAergic control within GP from a state dominated by local collateral inhibition that favor GP decorrelated firing mode[67,68] to a state driven by striatal inputs that favor GP neuronal correlations. The contribution of arkypallidal neurons, which send dense inhibitory projections onto striatal neurons[25,38], or the projections from a subpopulation of prototypic neurons that selectively contact striatal GABAergic interneurons[69] could further promote the amplification of $\beta$-synchronization[29].

Whether $\beta$-oscillations are an emerging property of the striatum[27], the striatal-GP microcircuit, or a larger feedback loop that includes the thalamus[70] is not known but will be important to address in future experiments. Another important aspect would be to determine if similar circuit mechanisms apply to the generation of abnormal $\beta$-synchronization in PD primates. Indeed, correlation analyses have shown that $\beta$-synchronization in primate GP seems to be principally driven by STN and not striatal activity[35]. In addition, the STN appears to be necessary for the expression of $\beta$-oscillations in MPTP-treated monkeys[32,71]. The contribution of the striatum is less clear: blockage of GABA transmission in the GP does not affect/suppress $\beta$-oscillatory

activity[32,71] and, despite the reduction of GP activity in MPTP primate as compared with control animals[35], there is still some debate as to whether striatal activity is increased[72] or not[35]. Altogether, our data draws strong and testable predictions that might help in resolving these issues. Indeed, if STN activity drives $\beta$-synchronization in primates, then STN activity should have an 'in-phase' relationship with the main population of GP neurons (i.e., the prototypic). We show here however that the scenario leading to the generation of abnormal $\beta$-oscillations dynamics in PD rodents is different and principally relies on inhibitory mechanisms that are orchestrated and broadcasted to the CBG network by GP neurons.

## Methods

**Animals**. All experimental procedures were performed on adult male Sprague-Dawley rats (9–21 weeks, Janvier Labs) in accordance with the European legislation for the protection of animals used for scientific purposes (directive 2010/63/EU). This project was also approved by the French ministry of higher education and research and the ethical committee of CNRS, Aquitaine Region (accreditation number 5012079-A). Rats were housed collectively (at least by two) per cage under artificial conditions of light (light/dark cycle, light on at 7:00 a.m.), temperature (24 °C), and humidity (45%) with food and water available ad libitum. A total number of 93 rats were used for this study of which 20 rats were excluded due to the low transduction rate of STN neurons as verified with electrophysiological recordings in STN (see 'Functional optogenetic mapping of laser effect in STN' for more details).

**Stereotaxic injection of 6-OHDA**. Unilateral 6-OHDA lesions were induced as previously described[10]. Briefly, rats (300–350 g) were anesthetized with isoflurane (induction/maintenance: 3–5/1.5%; Iso-vet®, Piramal healthcare), fixed on a stereotaxic frame (Unimécanique, M2e) and placed on heating blanket. An ophthalmic ointment (Liposic®, Bauch & Lomb Swiss) was used all along the surgery to prevent dehydration. Then, the rats received desipramine hydrochloride (25 mg/Kg, i.p; CAT#D3900, Sigma-Aldrich) 30 min before 6-OHDA injection in the right medial forebrain bundle (AP, −3.8 mm caudal to bregma; ML, 1.6 mm from midline). This pretreatment was used to prevent noradrenergic system lesion. After subcutaneous injection of xylocaine (15 mg/Kg; AstraZeneca), skull was exposed and a craniotomy was performed to inject 2.5 µL of 6-OHDA 3% (w/v; CAT#162957, Sigma-Aldrich) solution. This latter was dissolved in saline solution containing ascorbic acid 0.1% (w/v; CAT#A4544, Sigma-Aldrich) and was injected in five points (500 nL each) separated by 250 µm vertical intervals (−7 to −8 mm from cortical surface) with glass capillary (tip diameter 35 µm; 1–5 µL Hirschmann® microcapillary pipette, CAT#Z611239, Sigma-Aldrich) connected to Picospritzer® (Parker Hannifin). At the end of surgery, 5.5% glucose solution (w/v; CAT#G8270, Sigma-Aldrich) and buprenorphine (0.05 mg/Kg; Axience) were injected subcutaneously. To assess the 6-OHDA lesion, an apomorphine challenge was performed (0.05 mg/kg, s.c.; Apokinon®, Aguettant) 14–15 days after surgery[24,33]. Only rats performing 80 net rotations in 20 min were selected in this study.

**Stereotaxic injection of viral vectors**. All the virus used for our optogenetic manipulations were adeno-associated virus directly purchased from a vector core (UNC vector core) and microinjected under stereotaxic condition in control or hemi-lesioned rats using glass capillary (tip diameter 35 µm; 1–5 µL Hirschmann® microcapillary pipette, CAT#Z611239, Sigma-Aldrich) connected to a Picospritzer® pressure system (Parker Hannifin). Experiments of mCx optogenetic inhibition, were achieved using an AAV2/5-CamKII(calcium/calmodulin-dependent protein kinase II)-eArchT3-YFP viral vector ($3 \times 10^{12}$ viral particles/mL) injected in rats ($n = 13$, total volume of 2 µL) in four points into both primary (M1) and secondary (M2) mCx ipsilateral to the lesion (500 nL per injection sites at coordinates: AP, +4 mm rostral to bregma, ML, +2 and +2.8 mm from midline; AP, +3.4 mm rostral to bregma, ML, +1.8 and +2.6 from midline; DV, −1.5 mm from cortical surface). For STN manipulations, we injected in hemi-lesioned (ipsilateral to lesion) or normal animals (right hemisphere) the viral solutions AAV2/5-CamKII-eArchT3-YFP ($n = 13$ hemi-lesioned rats, $3 \times 10^{12}$ viral particles/mL), AAV5-hSyn-eArch3-YFP ($n = 9$ hemi-lesioned rats, $3 \times 10^{12}$ viral particles/mL), or AAV2/5-CamKII-hChR2(H134R)-YFP ($n = 7$ and 10 for hemi-lesioned and normal animals, respectively, $6.2 \times 10^{12}$ viral particles/mL) in two injection points in the STN (250 nL each, coordinates: AP, −3.6 and −3.9 mm caudal to bregma, ML, +2.5 mm from midline, DV, −7.5 mm from cortical surface). For GP optogenetic inhibition, we used an AAV2/5-hSyn-eArch3-YFP ($3 \times 10^{12}$ viral particles/mL) injected in two points with a volume of 300 nL at coordinates: AP, −0.9 mm caudal to bregma; ML, +2.8 mm from midline; DV, −5.6 and 6.1 mm from cortical surface in GP of hemi-lesioned ($n = 15$, ipsilateral to lesion) or control ($n = 18$, right hemisphere) animals.

**In vivo electrophysiological recording**. Electrophysiological recordings and optogenetic manipulations were realized at least 4 weeks after viral transduction under urethane anesthesia. Anesthesia was first induced with isoflurane (3–5%; Iso-vet®, Piramal healthcare), and maintained with urethane (i.p., 1.3 g/Kg.; CAT#U2500, Sigma-Aldrich). The animals were then secured in a stereotaxic frame (Unimécanique, M2e) and placed on heating blanket. After subcutaneous injection of xylocaine, skull was exposed and craniotomies were performed to unable electrophysiological recordings and optogenetic stimulation of the region of interest (same stereotaxic coordinates as above). In each rat, an ECoG of the sensorimotor cortex were realized (AP, +4.3 mm rostral to bregma; ML, +2 mm from midline) with a 1-mm screw juxtaposed to the dura mater[24,25]. All along the recording session, saline solution was used to prevent dehydration and anesthesia level was frequently controlled by examination of ECoG and by testing the response to light sensory stimuli (tall pinch).

In vivo extracellular recordings were performed using glass electrodes (1–3 µm tip end, 12–20 MΩ, GC150F, WPI) filled with a chloride solution 0.5 M containing neurobiotin tracer (1–2%, w/v; CAT#SP-1120, Vector laboratories) to perform juxtacellular labeling as described below[25]. For GP recordings, neurons were classified as prototypic and arkypallidal neurons based on their well-known electrophysiological signature across different brain state[25,38]. A subset of neurons ($n = 10$) were further labeled with neurobiotin and identified according to their molecular profiles: prototypic neurons expressed the transcription factor Nkx2.1 whereas arkypallidal neurons expressed the transcription factor FoxP2. Optogenetic manipulation of the recorded neurons was performed using an opto-electrode that was homemade by gluing an optical fiber (multimode Fiber, 0.22 NA, core diameter: 105 µm, Thorlabs) 1 mm and/or 2 mm above the tip of the glass recording electrodes. This method allows extracellular recording during opto-manipulation and juxtacellular labeling. Once the opto-electrode was implanted in the recording structure, the signal was amplified tenfold using an axoClamp 2B (in bridge mode; Molecular Devices). The recorded signal was then amplified 1000-fold and filtered using a differential AC amplifier (spike unit: 0.3–10 kHz, LFP: 0.1 Hz–10 kHz; model 1700, A-M Systems). For ECoG recording, the signal was amplified 1000-fold and filtered with the same differential AC amplifier (0.1 Hz–5 kHz). Finally, all recorded signals were digitalized at 20 kHz, by using the Power1401-3 connected to a computer equipped with Spike2 software (Cambridge Electronic Design). To facilitate the identification of STN during the recording sessions, a concentric bipolar electrode (SNEX-100, Rhodes Medical Instruments) connected to an isolated stimulator (DS3 Isolated Current Stimulator, Digitimer), was inserted in the mCx (coordinates: AP, +3.5 mm rostral to bregma, ML, +3.5 mm from midline, DV, −1.8 mm from cortical surface). STN was identified by checking online the STN short-latency triphasic response to mCx electrical stimulation. We then validated the recording location with histological control, in particular the location of the juxtacellularly labeled neurons and the track marks left by the opto-electrode.

**Functional optogenetic mapping of laser effect in STN**. In order to validate the results of our STN opto-inhibition and STN opto-excitation experiments performed in 6-OHDA rats, it was critical to verify functionally that the overall level of optical control achieved in STN was acceptable. For this reason, we adopted an opto-electrode mapping strategy that consisted of combining laser stimulation with STN extracellular recordings at multiple locations in the STN (i.e., at least three penetrations spaced by 150 µm in the antero-posterior or medio-lateral axis of the STN). In addition, only animals that had at least ten STN cells opto-tested in vivo and presenting >60% of significantly laser-modulated cells were kept for further analysis. Using these criteria, hemi-lesioned animals with poor STN optogenetic control were excluded from the study, that is: 9 out of 13 rats injected with AAV5-CamKII-eArch3-EYFP; 5 out of 9 injected with AAV5-hSyn-eArch3-EYFP; and 2 out of 7 injected with AAV5-CamKII-ChR2-EYFP. The level of STN optical control was also verified/quantified in normal animals injected with an AAV5-CamKII-ChR2-EYFP (four out of ten excluded) or with the AAV5-CamKII-eArchT3-EYFP (none excluded out of four).

**Optogenetic stimulation**. To apply optogenetic stimulation, we implanted optical fibers (multimode Fiber, 0.22 NA, core diameter: 105 µm, Thorlabs) into the GP (coordinates: AP, −0.9 mm caudal to bregma, ML, +2.8 mm from midline, DV, −5.6 and 6.1 mm from cortical surface) or STN (coordinates: AP, −3.7 mm to bregma, ML, +2.5 mm from midline, DV, −7.0 mm from cortical surface) or mCx. Note that for mCx inhibition, two optic fibers were implanted to maximize the volume of cortical inhibition (coordinates: AP, +4 mm rostral to bregma, ML, +2.2 mm from midline; AP, +3.4 mm rostral to bregma, ML, +2.2 mm from midline; DV, −1.0 mm from cortical surface). The power at the tip of the optics fibers was always measured with a power meter (PM100D, Thorlabs) right before insertion in brain tissue. Two optogenetic stimulation protocols were used in this study. The first one consisted of 2 s of continuous yellow or blue light (15 or 2.5 mW at the tip of optical fiber, respectively) every 10 s. The 2 s preceding the light pulse was used as baseline (OFF period) to compare the effect of the 2 s of optogenetic manipulation (ON epoch). Only recordings that contained a minimum of 30 laser stimulations (and at least 60 s during OFF and ON epochs) were kept for analyses to quantify the effect of the optogenetic inhibitions on $\beta$-synchronization (phase locking, ECoG power, and coherence). For each animal, the best

$\beta$-oscillations recording was determined based on the highest peak in the $\beta$-AUC (12–30 Hz) power spectrum calculation. The effect of our optogenetic manipulations on the neuronal firing rate measured in STN and GP neurons (in hemilesioned and control animals) were determined using at least ten laser stimulations. To reintroduce abnormal $\beta$-synchronization in control animal, we used a different protocol that consisted of 20 s of sinusoidal yellow or blue light stimulation (15 or 2.5 mW at the tip of optical fiber, respectively). Only the positive part of the sinusoid drove laser activation. The baseline OFF epoch consisted of the 20 s before light stimulation and was compared with the ON epoch that contains 20 s of sinusoidal stimulation. Only recording with at least three sinusoidal protocols (60 s) were included in this study. The laser diode (Errol laser) was controlled by an analogic signals sent by a Power1401-3 (Cambridge Electronic Design) and controlled with the Spike2 software (Cambridge Electronic Design).

**Decortication experiments and STN electrolytic lesion.** To test the importance of other cortical area in the generation mechanism of $\beta$-oscillations we performed large-scale decortication experiments. Removal of the temporal and frontal bones exposed most of the dorsal cortical surface. We removed the cortical tissue using an electrosurgical cautery (BC 50D, Anhui Yingte Electronic) and the depth of the lesion was assesses based on the visual inspection of the corpus callosum. As shown in Supplementary Fig. 2, this approach allowed to lesion a broad cortical volume including the M1, M2, S1, and a large part of the S2 cortex. In these experiments, brain-state activity was monitored through an ECoG recorded above mCx on the contralateral side of the lesion and combined with GP LFP recording (ipsilateral to lesion) using 32-channel silicon probe microelectrodes (Cambridge Neurotech, GB) in anaesthetized hemi-lesioned rats ($n = 3$).

To further test the contribution of STN in generating $\beta$-oscillations, we performed an electrolytic lesion of the STN in anaesthetized hemi-lesioned rats ($n = 3$). We first performed electrophysiological recordings to map the exact stereotaxic location and depth of the STN. We then lowered a concentric bipolar electrode (SNEX-100, Rhodes Medical Instruments) and located the tip of the electrode at the bottom of the STN. The electrode was connected to an isolated stimulator (DS3 Isolated Current Stimulator, Digitimer) and an electrolytic lesion was performed by applying a continuous current (3 mA during 20 s) in two stereotaxic coordinates separated by 300 μm in the rostro-caudal axis. Ipsilateral ECoG recording was performed before and after STN lesion.

**Juxtacellular stimulation.** In this study, we performed juxtacellular labeling to identify the location/cell-type of the recorded unit in STN or in GP. Briefly, the recorded unit was stimulated by a current pulse of 250 ms long (50% duty cycle, 1–10 nA) sent through the recording electrode by axoClamp 2B (in bridge mode; Molecular Devices). Only recording with at least 250 stimulations were included in this study.

**Data processing and analysis.** We performed simultaneous recording of the mCx ECoG and STN unit activity at a sampling rate of 20 kHz during epoch containing $\beta$-synchronization. All data processing (spike sorting and ECoG filtering) were realized offline with Spike2 software (Cambridge Electronic Design). The ECoG signal was high-pass filtered at 0.5 Hz (DC remove) and low-pass filtered at 500 Hz following by a down-sampling at the same frequency (to avoid aliasing effect) for further field analysis. The frequency resolution was set at around 1 Hz (FFT size = 500/512 Hz) to analyze the root mean square ECoG power spectrum or to calculate the coherence between spike train and ECoG (using 'coher' script, freely available on CED website). For generating peri-event spectrograms, we computed in Matlab® R2016a (MathWorks, Natick, MA, USA) using the ft_freqanalysis function (time/ frequency slide windows: 50 ms/0.071 Hz, cycle number: 7) of the Fieldtrip toolbox[73]. Peri-stimulus time histograms or PSTH (width: 6 s, offset: 2 s, bin size: 50 ms) of each spike trains were generated using spike2 and a custom Matlab® script. Both frequency and count PSTH were generated for each spike train to evaluate the effect of the optogenetic manipulations. A response was classified as statistically significant if the spike count values of three consecutive bins within the first 400 ms of light pulse delivery were <−2SD (inhibitory response) or >2SD (excitatory response). SD was measured during OFF epoch. We also analyzed in spike2 the frequency of each spike train during OFF and ON epoch. This allowed us to calculate the MI for evaluating the frequency increasing induced by our manipulations. MI was calculated as follow: (frequency ON − frequency OFF)/(frequency ON + frequency OFF).

To generate the linear phase histograms in spike2 (bin size: 10°), we detected the peak of $\beta$-oscillations in the band-pass filtered (12–30 Hz) ECoG using a threshold detection as previously described[24,25]. The linear phase histograms were then transferred to Matlab® for further analysis using circular statistics toolbox[74]. To determine if a spike activity was significantly modulated by the $\beta$-oscillations measured in the ECoG, we performed the Rayleigh test for circular uniformity using the 'circ_rtest' function which evaluates if the spike unit activity is uniformly distributed across the ECoG $\beta$-oscillation cycles (H0, $p > 0.05$). If H0 was rejected, neuron was considered as significantly entrained by $\beta$-oscillations. The mean angle and vector length for each spike train were also measured by using the 'circ_mean' and 'circ_r' function, respectively. Finally, the mean angles of entrained neurons were compared (OFF vs. ON epoch or between groups) by using the Watson–Williams $F$ test ('circ_wwtest' function, circular statistics toolbox).

**Random spike removing in mCx experiment.** Our mCx optogenetic inhibition was associated with a reduction in the firing rate of STN neurons and a consequent decrease in the coherence value measured between STN spike and ECoG reduction in the functional connectivity. To determine the influence of change in firing rate on the spike-ECoG coherence value we performed random spike removal on our STN spike timestamps using the 'randsample' Matlab® function. We analyzed the effect of a 20 or 50% decrease in STN firing rate and performed coherence analysis in Spike2 as previously described.

**Effect of optogenetic stimulation across different $\beta$-power.** To analyze the effect of the optogenetic stimulation across the $\beta$-power distribution, we compared the root mean square power of every individual $\beta$-oscillations period as measured through the AUC within the 12–30 Hz frequency band for each stimulation epoch (OFF laser vs. ON laser) for each neuron. The $\beta$-AUC was calculated in Matlab® through an interpolation method ('interp1' function) of the power spectrum in the 12–30 Hz frequency band. A $\beta$-AUC difference ($\beta$-AUC above interpolated line) − ($\beta$-AUC below interpolated line) was calculated for every 2 s long stimulation epoch (OFF and ON laser stimulation). Finally, the cumulative distribution was computed with the 'cdfplot' function (Matlab®).

**Beta bursts detection.** The $\beta$ bursts were extracted from the ECoG signal recorded above the mCx and according to a threshold detection method (See Supplementary Fig. 1a). The signal was first down sampled at 500 Hz and band-pass filtered in the $\beta$-band (see 'Data processing and analysis'). The filtered EcoG signal was then imported into Matlab and the envelope of the $\beta$ signal was calculated using the hilbert function. The signal was defined as a beta burst by (1) the amplitude of the envelope exceeding the 75th percentile threshold of the analytic signal measured during the entire recording, and (2) the duration of the burst was longer than 50 ms. Finally, the duration and the number of $\beta$ bursts were averaged accordingly to the three laser epochs (i.e., Before, OFF and ON) and statistically compared.

**Tissue processing and histological control.** At this end of recording day, the rats were sacrificed with an overdose of pentobarbital sodium (150 mg/Kg, i.p.; Axience). An intracardiac perfusion (PBS 0.01 mM following by formaldehyde 4%) was then performed for further histological validations. The brain was kept overnight in a solution of PBS 0.01 mM/formaldehyde 4% (v/v; CAT#20909.330, VWR) and then cut in 50-μm slices with a vibratome (VT1000, Leica Microsystems). To reveal the juxtacellularly labeled neurons, the slices were incubated overnight in a solution of PBS 0.01 mM/Triton™ X100 0.3% (v/v; CAT#T9284, Sigma-Aldrich) containing streptavidine-CY3 Zymax (1/1000, v/v; CAT#438315, Life Technologies). The slices were then washed in PBS 0.01 mM before mounting onto slides in vectashield medium (CAT#H-1000, Vector laboratories). Additional staining was performed through indirect immunofluorescence staining using primary and fluorescent secondary antibodies. For all immunostainings, the slices were incubated overnight in a solution of PBS 0.01 mM/Triton™ X100 0.3% (v/v) containing the primary antibody. The slices were then washed three times in PBS 0.01 mM, incubated 4 h in a solution of PBS 0.01 mM/Triton™ X100 0.3% (v/v) containing the secondary antibody, washed again in PBS 0.01 mM and mounted onto slides in vectashield medium (CAT#H-1000, Vector laboratories). The primary antibodies used in this study were: chicken anti-YFP (1/1000, v/v; CAT#GFP-1020, Avès labs), rabbit anti-Nkx2.1 (or anti-TTF-1, prototypic neuron labeling, 1/500, v/v; CAT#H-190, Santa Cruz), and goat anti-FoxP2 (arkypallidal neuron labeling, 1/500, v/v; CAT#N-16, Santa Cruz). The secondary antibodies used were: Alexa Fluor® 488 donkey anti-chicken (1/500, v/v; CAT#703-545-155, Jackson ImmunoResearch), CY™5-conjugated donkey anti-rabbit (1/500, v/v; CAT#711-175-152, Jackson ImmunoResearch), and Brilliant Violet™421-conjugated donkey anti-goat (1/500, v/v; CAT#705-675-147, Jackson ImmunoResearch). All these primary and secondary antibodies have been characterized and used in previous similar work[39]. The images were acquired with a fluorescence microscope (Axio Imager 2, Zeiss) or a confocal microscope (Leica, SP8) and were analyzed with Fiji (ImageJ 1.52). To determine if our opto-mapping recordings applied to STN opto-inhibition experiments in parkinsonian rats accurately capture the level of STN infection (Supplementary Fig. 3a, b), we performed a histological quantification of the effectiveness and spread of virus infection in the STN. To do so, we determined the intensity of STN fluorescence in the EYFP channel using region of interest (ROI) measures in Fiji. Briefly, we mounted serial brain sections spaced by 300 μm and containing STN sagittal slices at different medio-lateral coordinate (i.e., 2.9/2.6/2.3/ 2 mm from the midline). For each slices, the value of ROI was normalized according to the background labeling in a region devoid of EYFP signal and taken just below the STN region. For each rat, we averaged the different STN ROI values and correlated this measure to the percentage of opto-inhibition as determined by our opto-mapping strategy.

**Statistical analysis.** The statistical analyses were two-tailed statistical tests with a risk $\alpha$ set at 0.05 and were performed in SigmaPlot 12 (Systat Software). For independent samples, we applied the normality (Shapiro–Wilk test) and equal variance tests. A $t$-test was used if the distributions were normal and the group variances were equal. Otherwise, the Mann–Whitney signed rank test was used. In a same way, Kruskal–Wallis one-way ANOVA was used instead of one-way

ANOVA, when normality distributions and variance homogeneity were not verified. ANOVA was followed by Student–Newman–Keuls post hoc test. For dependant sample, the paired *t*-test was used excepts if the normality distribution test failed (Shapiro–Wilk test, $p < 0.05$). In the latter case, the Wilcoxon signed rank test was used. In same way, Friedman repeated measures ANOVA on ranks was used instead of one-way repeated measure ANOVA, when normality distribution was not verified. ANOVA repeated measure was followed by Holm–Sidak (parametric) or Dunn's (nonparametric) post hoc test.

**Reporting summary**. Further information on research design is available in the Nature Research Reporting Summary linked to this article.

## Data availability
The full raw data that support the findings of this study are available from the corresponding author upon reasonable request. The source data underlying Figs. 1c, e, f, h–j, o, 2e, f, h, i, 3c–f, 4d–f, h–j, l–n, 5c–e, i–j, 6c–g and Supplementary Figs. 1b–g, i–j, 2b, 3a–d, 4b, 5c–h, 6a–e, 7a–j, 8a–h, 9a, b, d, e are provided as a Source Data file.

## Code availability
The codes for the analysis used in this study are available from the corresponding author upon reasonable request.

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

## Acknowledgements

This work was supported by grants from the French Agence Nationale de la Recherche (ANR-14-CE13-0024-01 and ANR-15-CE37-0006), from the CNRS PEPS Idex Bordeaux (UB101 CR-2014R), and the LABEX BRAIN ANR-10-LABX-43. B.C. was supported by a fellowship from French Ministry (Higher Education, Research and Innovation), from France Parkinson nonprofit organization (UB320 CR-3219R), and from the LABEX BRAIN (UB106 CR-3219R). We are grateful to T. Nguyen and H. Orignac for technical assistance, Drs D. Hansel, and P. Magill for insightful scientific discussions; Drs M. Deffains, J. Baufreton, and A. McDonald for comments on the paper.

## Author contributions

Conceptualization, B.C., T.B., and N.P.M.; methodology, B.C., A.A., and N.P.M.; formal analysis and software, B.C., A.L., and N.P.M.; investigation, B.C., A.A., S.E., and N.P.M.; writing—original draft, B.C.; writing—review and editing, N.P.M.; funding acquisition, N.P.M.; resources and supervision, N.P.M.

## Competing interests

The authors declare no competing interests.
