## [Peer Review File · Nature Communications]

Reviewers' comments:

Reviewer #1 (Remarks to the Author):

The authors perform a detailed series of optogenetic manipulations in motor cortex, STN, and GP of rodents under urethane anesthesia. Most experiments are done on 6-OHDA lesioned rats (with an inclusion criteria specifying degree of parkinsonism in rotometer tests). A small number are done on normal rats. The primary results are that, in the parkinsonian state, neither cortical nor STN optogenetic interventions disrupt oscillatory dynamics in the other structures of the network; while pallidal inactivation does. Strengths of the paper are the systematic examination of three different key locations in the basal-ganglia- thalamocortical circuit, the thoroughness of control experiments and the framing of results in terms of recent theoretical advances in rodent basal ganglia physiology, such as the distinction between prototypic and arypallidal cells and their phasic interactions with other basal ganglia nuclei. It is a technical "tour de force" and included many difficult experiments that were clearly done with great care.

Several elements of this paper may make it appropriate for a specialized audience who focuses particularly on rodent basal ganglia circuitry, but a bit less so for a wider audience. First, all animals are anesthetized so there are no behavioral correlates. Thus it is not clear if any of the manipulations alter parkinsonian motor function. If so, then it would be possible to associate, or dissociate, particular network manipulations with/from the occurrence of motor abnormalities, which would greatly increase the general interest of the paper. Also, it is not clear if the network oscillations described are present in the awake state (or only under urethane anesthesia (as is suggested by prior studies).

Although the authors examine "beta dynamics" by examining power spectra over relatively short intervals, they do not specifically address the effect of network manipulations on beta burst dynamics. Shortening of beta bursts seem to be key to improving Parkinsonian motor signs (based on several recent papers from Peter Brown's group). It is not clear if their "area under the curve" method of quantifying beta band oscillations, even if performed on short data segments, is informative with respect to beta bursting properties; and the link to beta burst length is not made.

The description of oscillatory activity as abnormal would depend on showing it to be absent in the normal animals under similar anesthetic states; prior work suggests this is the case; and the section on driving STN neurons in the normal state obliquely addresses the physiological differences between normal and parkinsonian, but the authors should more clearly state the effects of induction of parkinsonism in their model, referring to prior papers on this if necessary.

The discussion is dense and relatively hard to follow for a reader not steeped in the details of rodent basal ganglia microcircuitry. It would help if the work were contextualized for a wider audience.

Reviewer #2 (Remarks to the Author):

This paper seeks to answer a fundamental, longstanding question in the field of PD – what is the locus of beta oscillation generation? To interrogate this question, the authors perform a set of careful optogenetic experiments to silence potential sources of beta oscillations in dopamine depleted rats. They also generate beta oscillations in the same regions to probe their similarity to Parkinsonian oscillations as they propagate through the basal ganglia. This paper makes some strong points that will add valuable and novel information to the Parkinsonian oscillation literature – namely, that motor cortex is not necessary for beta oscillations while GP is. In particular, they carefully show that suppression specifically of GP, and not similar interventions, quenches beta oscillations. Overall, this paper makes an important contribution to the field of Parkinsonian

oscillations. I think it will be rigorous, impactful and of broad interest.

Major points:

1. Figure S2: The authors should label the PSDs in Figure S2B, as this could greatly change the figure's interpretation. If the green is when M1 is lesioned, it would appear that lesioning M1 does, in fact, eliminate beta oscillations.
2. Figure 3, 6: The authors show alternative evidence that STN is not (and GP is) responsible for beta oscillation generation by generating artificial beta oscillations in these nuclei and comparing the oscillations to those observed in the depleted state. These are very interesting experiments, but should be interpreted with a bit of caution:
 - a. An inhibitory rhodopsin was used to generate beta oscillations in GP, citing increased striatal inhibition onto GP neurons following depletion. In contrast, an excitatory rhodopsin was used to simulate beta oscillations in STN. But if their data suggests M1 is not the source, then inhibitory inputs from the GPe might be more important to mimic, requiring an inhibitory opsin.
 - b. An argument against the STN as the source of beta is because artificially-induced STN beta increases firing rate of prototypic GP neurons, whose firing rates are normally decreased in the depleted state. But this result could be explained because they are using ChR2 to excite the STN, which by extension, excites the GP. The authors should show that their results are robust regardless of which rhodopsin is used in the areas.
 - c. It is unclear how the authors chose the measurements that they considered proof that an oscillation was DD-like and which measurements were irrelevant. For instance, GP stimulation mimics downstream firing rate changes and phase locking values seen in DD, while STN stimulation does not. However, both increase the coherence of the downstream region (GP for STN and vice-versa) with motor cortex, and only STN stimulation is able to recapitulate the antiphase relationship between prototypic and arkyvallidal GP neurons.
3. Figure 2: The authors excluded animals in which fewer than 60% of neurons were negatively modulated. This is rather close to the 75% average of those animals which remained in the experiment after this thresholding. Since the authors only recorded from an average of about 15 neurons per animal (inferred from table S2), only a small change in the sampled neurons would move an animal from the average of the included group to below the cutoff threshold. This raises the possibility that the 75% value mentioned could be an overestimate of the effectiveness of their inhibition. The authors should provide histological analysis of the viral spread across all animals and its correlation with the perceived effectiveness of inhibition.
4. Despite the valiant efforts of the authors to inhibit the STN, including two different inhibitory opsins, there are lingering concerns about how effectively the STN is silenced. These concerns could be alleviated somewhat by recordings from more units across the full extent of the STN. But it might be necessary to fully lesion (or silence with muscimol?) the STN, as the authors did for M1 earlier in the paper. Alternatively, perhaps infusing glutamate blockers into the GPe could be used to more completely block input from the STN. While non-optogenetic options may be less precise due to spread to axons and terminals not originating from STN, a more extreme manipulation that still fails to affect beta oscillations would be a wise way to prove this negative result. If none of these experiments work, I would still be willing to consider the paper for publication, as it is a heroic body of work. But, because of the surprising nature of the result, and concerns about the effectiveness of STN stimulation, additional experiments would bolster the paper's conclusions.

Minor points

1. Figure 1M: The meaning of the bottom panel is hard to understand – can it be explained better in the figure legend and/or main text?
2. Figures 2H and 4M: These appear to be the schematic of 1M redisplayed exactly – is this necessary to be in all three figures?
3. Methods, Data Processing and Analysis: "A response was classified as statistically significant if

the spike count values of 3 consecutive bins within the first 400 ms of light pulse delivery were $< -2SD$..." Since the bin size is 50msec, this means the authors are testing bins $\{1,2,3\}$; $\{2,3,4\}$... $\{6,7,8\}$ which is six comparisons. The authors should correct for these multiple comparisons or state that they have already done so. In the opposite direction, the authors are probably making these statistics more stringent than necessary by requiring that three bins each be statistically significant, though it is not clear by how much. Theoretically, $p < 0.05$ actually has a false positive rate of $(0.05)^3$ when three bins are required, but since consecutive bins are not independent, the false positive rate is somewhere between $(0.05)^3$ and 0.05. The authors could consider simply using a larger bin size (while still correcting for multiple comparisons).

4. Line 168: typo, reproduced should be reproduce.

5. Line 362: typo, compare should be compared.

Response to Reviewers

We would like to take the opportunity to thank the reviewers for taking the time to analyze our work and for their constructive comments and feedbacks which have helped us to improve the quality of our manuscript. We are grateful for the opportunity to resubmit our work and we hope that you will find our revisions to the below comments acceptable.

The revised version of our manuscript includes new analysis of electrophysiological (i.e. burst analysis) and histological data (i.e. quantification of STN virus infection). We also performed additional experiments: 1/ We further strengthen our results showing the lack of effect induced by our STN opto-inhibition on abnormal β -oscillations generation by carrying out STN lesioning in 6-OHDA lesioned animals. We opted for an electrolytic lesion of the STN and were able to show that the expression of β -oscillations in parkinsonian animals (n=3) was not affected by this extreme STN manipulation. 2/ We also were able to confirm that our previous results obtain when artificially reintroduction β -oscillations using a ChR2 excitatory approach in the STN of normal rats is indeed robust regardless of the rhodopsin used. This has been achieved by performing additional experiments using an ArchT inhibitory approach in the STN of normal rats (n=4). Similar to the ChR2 experiments, we found that these ArchT opto-patterning experiments do not recapitulate the functional properties (i.e. the correct STN/GP spike-timing relationship and firing rate changes) of parkinsonian β -oscillations. In conclusion, the results obtain from the ChR2-excitatory and ArchT-inhibitory manipulations of the STN clearly support the view that STN cannot be the driver of the abnormal network changes present in cortico-basal ganglia circuits during parkinsonism. We sincerely believe that, with these novel data, the conclusion of our study is substantially strengthened. All the changes and text addition to the previous version of our manuscript have been highlighted in red in this revised version. Please, find below our point to point response to the reviewers comments.

Reviewer #1:

The authors perform a detailed series of optogenetic manipulations in motor cortex, STN, and GP of rodents under urethane anesthesia. Most experiments are done on 6-OHDA lesioned rats (with an inclusion criteria specifying degree of parkinsonism in rotometer tests). A small number are done on normal rats. The primary results are that, in the parkinsonian state, neither cortical nor STN optogenetic interventions disrupt oscillatory dynamics in the other structures of the network; while pallidal inactivation does. Strength of the paper are the systematic examination of three different key locations in the basal-ganglia- thalamocortical circuit, the thoroughness of control experiments and the framing of results in terms of recent theoretical advances in rodent basal ganglia physiology, such as the distinction between prototypic and arky pallidal cells and their phasic interactions with other basal ganglia nuclei. It is a technical “tour de force” and included many difficult experiments that were clearly done with great care.

Several elements of this paper may make it appropriate for a specialized audience who focuses particularly on rodent basal ganglia circuitry, but a bit less so for a wider audience. First, all animals are anesthetized so there are no behavioral correlates. Thus it is not clear if any of the manipulations alter parkinsonian motor function. If so, then it would be possible to associate, or dissociate, particular network manipulations with/from the occurrence of motor abnormalities, which would greatly increase the general interest of the paper.

Response: In this work we have identified optogenetic manipulations that can either suppress the abnormal β -oscillations present in parkinsonian rats or, on the opposite, reintroduce artificial β -oscillations that closely mimic (or not) the functional properties of real parkinsonian β -oscillations.

We are, of course, very interested in determining how these circuit manipulations impact on movement generation in behaving rats. The primary goal of this paper, however, was to shed light on the neuronal circuitry involved in the generation of this abnormal network activities present in PD. This has never been assessed with optogenetic tools before this work. Being able to do so and provide new data challenging the conventional ideas on how β -oscillations are generated in these circuits represents a big accomplishment that, we believe, deserves publication by itself. In addition, to correctly dissect the behavioral correlates underlying the abnormal network dynamics present in parkinsonism, one needs to be able to account for the consequences induced by the change in the firing rate independently from the effect induced by the change in synchronization. This is currently not a trivial issue to overcome. Indeed, all our optogenetic network manipulations affect simultaneously the level of synchronization and the firing rate which might lead to erroneous conclusions on the behavioral consequences induced by the reintroduction of β -oscillations. In addition, we believe that this important question will be better addressed using mice (and not rats) since the access to optogenetic manipulation of specific neuronal population (i.e. striatopallidal vs. subthalamic nucleus neurons) is facilitated by the use of transgenic animals.

Also, it is not clear if the network oscillations described are present in the awake state (or only under urethane anesthesia (as is suggested by prior studies)).

Response: The presence of these network oscillations in awake vs. urethane anesthesia is an important issue and we have now added a sentence (line 160) and included the references that describe the brain-state dependency of β -oscillations in both awake or the anesthetized condition.

Although the authors examine “beta dynamics” by examining power spectra over relatively short intervals, they do not specifically address the effect of network manipulations on beta burst dynamics. Shortening of beta bursts seem to key to improving Parkinsonian motor signs (based on several recent papers from Peter Brown’s group). It is not clear if their “area under the curve” method of quantifying beta band oscillations, even if performed on short data segments, is informative with respect to beta bursting properties; and the link to beta burst length is not made.

Response: We thank the reviewer for this important comment and we fully agree that the calculation of the β -power as determined by the ‘area under the curve’ (AUC) does not provide a full picture of the dynamical properties of β -oscillations, especially its bursting properties. This has been amended in the revised version of our manuscript and we now provide the quantification for the β -burst (i.e. counts and duration) using similar approaches as the one developed by the group of Peter Brown and collaborators. These new analyses are presented in **Figures S1 A-C** for the mCx opto-inhibition, in **Figures S2 C, D** for STN opto-inhibition, and in **Figures S7 I, J** for GP opto-inhibition. In short, we found that these additional data agree with our previous conclusions that the mCx is not necessary for the genesis of β -oscillations, the STN activity only plays a supportive role, whereas the GP activity is critical for both the generation and the propagation mechanism.

The description of oscillatory activity as abnormal would depend on showing it to be absent in the normal animals under similar anesthetic states; prior work suggests this is the case; and the section on driving STN neurons in the normal state obliquely addresses the physiology differences between normal and parkinsonian, but the authors should more clearly state the effects of induction of parkinsonism in their model, referring to prior papers on this if necessary.

Response: As mentioned in our response of a previous comments, the expression of abnormal β -oscillations in parkinsonian rats has been described to be brain-state dependent. Indeed, our past work has clearly established that abnormal β -oscillations only emerge when the cortex is in the activated state but not when engage in slow-wave oscillation. This property has been described both in

anesthetized and awake lesioned rats. Interestingly, we have observed that the cortical detection (using ECoG measurements) of the artificially-induced β -oscillations in normal rats follows the same brain-state dependency rule and, as a consequence, our experiments were performed in the activated state to maximize the induction/propagation of β -oscillations. We do not have any explanations and did not characterize this phenomenon but these similarities between the natural vs. the artificial β -oscillations have now been added to the revised version of our manuscript (see line 157-162).

The discussion is dense and relatively hard to follow for a reader not steeped in the details of rodent basal ganglia microcircuitry. It would help if the work were contextualized for a wider audience.

Response: To help readers not familiar with rodent basal ganglia microcircuitry to contextualize our work, we added schematics that present the different hypothesis tested in this work. Due to space limitation, these schematics were added as supplementary Figures in the discussion (see Figure S10). We have also made minor cosmetic changes to the discussion; in particular it now includes a concluding paragraph discussing the circuit mechanisms of β -oscillations generation. It was however difficult to identify which part dealing with the microcircuitry should be simplified and we do think that the points discussed on the genesis of β -oscillations in the current version are very important to be discussed for the field.

Reviewer #2:

This paper seeks to answer a fundamental, longstanding question in the field of PD – what is the locus of beta oscillation generation? To interrogate this question, the authors perform a set of careful optogenetic experiments to silence potential sources of beta oscillations in dopamine depleted rats. They also generate beta oscillations in the same regions to probe their similarity to Parkinsonian oscillations as they propagate through the basal ganglia. This paper makes some strong points that will add valuable and novel information to the Parkinsonian oscillation literature – namely, that motor cortex is not necessary for beta oscillations while GP is. In particular, they carefully show that suppression specifically of GP, and not similar interventions, quenches beta oscillations. Overall, this paper makes an important contribution to the field of Parkinsonian oscillations. I think it will be rigorous, impactful and of broad interest.

Major points:

1. *Figure S2: The authors should label the PSDs in Figure S2B, as this could greatly change the figure's interpretation. If the green is when M1 is lesioned, it would appear that lesioning M1 does, in fact, eliminate beta oscillations.*

Response: We agree that the interpretation of Figure S2B could have been confusing in its old format. Figure S2 shows that despite removal of most of the cortex, β -oscillations are still abnormally present in basal ganglia regions as illustrated by the expression of a distinctive peak in the power spectrum within β frequency range (12-30Hz). To avoid any confusion in the interpretation of the figure, the display of Figure S2B has been changed and split in three different power spectrums (one power spectrum per decorticated rat).

2. *Figure 3, 6: The authors show alternative evidence that STN is not (and GP is) responsible for beta oscillation generation by generating artificial beta oscillations in these nuclei and comparing the oscillations to those observed in the depleted state. These are very interesting experiments, but should be interpreted with a bit of caution.*

a. *An inhibitory rhodopsin was used to generate beta oscillations in GP, citing increased striatal inhibition onto GP neurons following depletion. In contrast, an excitatory rhodopsin was used to*

simulate beta oscillations in STN. But if their data suggests M1 is not the source, then inhibitory inputs from the GPe might be more important to mimic, requiring an inhibitory opsin.

Response: The idea behind the use of an excitatory Chr2 opsin to simulate β -oscillations in STN was to best reproduce the known firing properties of STN neurons during parkinsonian β (in particular, the firing hyperactivity and the abnormal β -synchronizations). Another advantage of this excitatory approach is that it also provides insights on the causal contribution of the hyperdirect pathway (by directly mimicking this excitatory drive at the level of STN). Our data clearly shows that this excitatory β -patterning approach does not reproduce the classic features of parkinsonian β -oscillations whereas GPe inputs do. Considering this, it seems logical to propose that the use of an inhibitory opsin (rather than an excitatory one) in STN might best mimic the impact of GPe inputs and, therefore, best emulate parkinsonian β -oscillations. This is a good idea and we decided to test it using ArchT3.0 virus injected in the STN of normal rats (n=4). The results of these experiments are presented in a new supplementary figure (Figure S5) in our revised manuscript. We found that STN opto-inhibitory approach could generate abnormal β -oscillations in cortico-basal ganglia loop, but that it still does not reproduce the results obtain with GPe opto-inhibition. Indeed, STN opto-inhibition at β -frequency results in an overall reduction of STN firing which is different from the disinhibitory effect induced with GP opto-inhibition that rightfully lead to STN hyperactivity (as in parkinsonism). In addition, the rhythmic inhibition of STN neurons is perceived by GPe neurons as a rhythmic decrease in excitatory drive which causes prototypic neurons to phase lock their activity to the remaining STN spiking. This ‘in-phase’ STN/prototypic firing is different from the ‘antiphase’ firing relationship present during parkinsonian β -oscillations (and reproduced by GPe opto-inhibition). Interestingly though, STN opto-inhibition could accurately reproduce the opposition of phase between prototypic and arky pallidal neurons and this further illustrate that the interaction between prototypic and arky pallidal neurons is principally governs by disynaptic inhibition from prototypic cells. These additional data are now described in the results paragraph starting from line 140 to line 196.

b. An argument against the STN as the source of beta is because artificially-induced STN beta increases firing rate of prototypic GP neurons, whose firing rates are normally decreased in the depleted state. But this result could be explained because they are using Chr2 to excite the STN, which by extension, excites the GP. The authors should show that their results are robust regardless of which rhodopsin is used in the areas.

Response: Following this suggestion and as mentioned in our above reply, we performed additional ArchT opto-patterning experiment in STN (Figure S5). By doing so, we were able to show that the results obtain with STN opto-inhibition on the STN/GP spike-timing are similar to the one obtain with STN opto-excitation. The data collected with STN opto-manipulation are thus robust regardless of the opsin used in the STN. In summary, when our opto-manipulations solely impact on the activity of STN, we found that STN and prototypic neurons enter a state of ‘in-phase’ relationships. On the contrary, when the circuit manipulation is applied to the GP then this perturbation lead to an ‘antiphase’ STN/prototypic relationship.

c. It is unclear how the authors chose the measurements that they considered proof that an oscillation was DD-like and which measurements were irrelevant. For instance, GP stimulation mimics downstream firing rate changes and phase locking values seen in DD, while STN stimulation does not. However, both increase the coherence of the downstream region (GP for STN and vice-versa) with motor cortex, and only STN stimulation is able to recapitulate the antiphase relationship between prototypic and arky pallidal GP neurons.

Response: We agree with the reviewer that this is an important point to clarify. We principally focused our measurements both on the STN/GP neurons spike-timing relationships and their

respective firing rate changes. This is now explicitly stated in the revised version of our manuscript (line 168). Our goal was to reproduce the same or as many as possible features of the parkinsonian β -oscillations. It is true that the GP opto-inhibition does not correctly recapitulate the antiphase response of prototypic vs. arky pallidal neurons whereas STN stimulations do. We have added one sentence (line 295) to reflect on this interesting finding and we think it can be explained by the fact that our GP opto-inhibition approach is global rather than specific to a GP cell-type (prototypic vs. arky pallidal neurons).

3. Figure 2: The authors excluded animals in which fewer than 60% of neurons were negatively modulated. This is rather close to the 75% average of those animals which remained in the experiment after this thresholding. Since the authors only recorded from an average of about 15 neurons per animal (inferred from table S2), only a small change in the sampled neurons would move an animal from the average of the included group to below the cutoff threshold. This raises the possibility that the 75% value mentioned could be an overestimate of the effectiveness of their inhibition. The authors should provide histological analysis of the viral spread across all animals and its correlation with the perceived effectiveness of inhibition.

Response: We realized that, in the method, we did not make it sufficiently clear on how we performed our opto-electrode mapping in the STN. All our opto-mapping experiments were done by performing multiple penetrations in the rostro-caudal or medio-lateral axis of the STN. This is an important technical detail because it suggests that our perceived effectiveness of inhibition is not bias for one site of recording but rather a good global estimate, even if the number of recorded neurons per animal is (relatively) low. These methodological details have now been added in the method of the revised manuscript. What was interesting performing these opto-mapping experiments is that we could record STN neurons separated by tens of microns (during the same penetration) with opposite laser response. We think that this micro-domain heterogeneity of laser-responses is best captured using electrophysiology mapping as compare to global EYFP signal. We nevertheless performed a quantitative analysis of the EYFP fluorescent signal for all the Arch opto-experiments performed in 6-OHDA animals. The results of these analyses are described in Figure S3 A (for hSyn-Arch) and B (for CamKII-eArchT). It shows that there is a good correlation between the EYFP signal and the perceived effectiveness of inhibition. However, if the global EYFP signal can clearly separate the ‘missed’ from the ‘hit’ STN virus injection, we are not 100% confident that this quantitative histological approach can be used on its own to determine which animals should be ‘included’ or ‘excluded’, especially when their ‘perceived’ effectiveness of inhibition is close to 50%. However, a combination of both might be the best way.

4. Despite the valiant efforts of the authors to inhibit the STN, including two different inhibitory opsins, there are lingering concerns about how effectively the STN is silenced. These concerns could be alleviated somewhat by recordings from more units across the full extent of the STN. But it might be necessary to fully lesion (or silence with muscimol?) the STN, as the authors did for M1 earlier in the paper. Alternatively, perhaps infusing glutamate blockers into the GPe could be used to more completely block input from the STN. While non-optogenetic options may be less precise due to spread to axons and terminals not originating from STN, a more extreme manipulation that still fails to affect beta oscillations would be a wise way to prove this negative result. If none of these experiments work, I would still be willing to consider the paper for publication, as it is a heroic body of work. But, because of the surprising nature of the result, and concerns about the effectiveness of STN stimulation, additional experiments would bolster the paper’s conclusions.

Response: We agree with the reviewer comment and although the results of our STN opto-inhibition in parkinsonian rats and the STN opto-excitation in normal animals converge toward the idea that

STN was not driving the oscillations, we were missing the key ‘removal’ experiments in our previous study. To avoid any problem in interpreting a negative result caused by the infusion of a drug in the STN (i.e. genuine lack of effect vs. bad diffusion vs. problem of concentration) and to match with what we had done previously for the decortication, we opted for an electrolytic lesion of the STN induced by current injection. The revised version of our manuscript now includes the consequence of STN lesion performed in three 6-OHDA-lesioned rats. The results of these experiments are now presented in Figure S4. In agreement with our optogenetic experiments, we found that the full lesion of STN has no effect on the expression of abnormal β -oscillations.

Minor points:

1. *Figure 1M: The meaning of the bottom panel is hard to understand – can it be explained better in the figure legend and/or main text?*

Response: We changed the display of Figure 1M to better illustrate our strategy as well as slightly change the phrasing in the legend. Hopefully, these modifications will facilitate the understanding of the panel.

2. *Figures 2H and 4M: These appear to be the schematic of 1M redisplayed exactly – is this necessary to be in all three figures?*

Response: We agree that the schematic of 1M was redundant in Figures 2H and 4M. It has been modified in the new version.

3. *Methods, Data Processing and Analysis: "A response was classified as statistically significant if the spike count values of 3 consecutive bins within the first 400 ms of light pulse delivery were $< -2SD$..." Since the bin size is 50msec, this means the authors are testing bins {1,2,3}; {2,3,4} ... {6,7,8} which is six comparisons. The authors should correct for these multiple comparisons or state that they have already done so. In the opposite direction, the authors are probably making these statistics more stringent than necessary by requiring that three bins each be statistically significant, though it is not clear by how much. Theoretically, $p < 0.05$ actually has a false positive rate of $(0.05)^3$ when three bins are required, but since consecutive bins are not independent, the false positive rate is somewhere between $(0.05)^3$ and 0.05. The authors could consider simply using a larger bin size (while still correcting for multiple comparisons).*

Response: This criterion for detecting a significant change in the mean level of activity if it goes above or below the 2 SD thresholds is rather standard when analyzing the neuronal response to a stimulus (Martiros et al., 2018; Mathai et al., 2015; Nambu et al., 2000). As the reviewer mentioned, each bin laying outside the 2SD has a p value of 0.05. If one assumes a Poisson distribution of the spike trains, time bins are independent and 3 consecutive bins will have a p value of $(0.05)^3$. Correcting for multiple comparisons, this leads to $p = (0.05)^3 * 8 = 0.001$. This indicates that our detection method has a p value of 0.001 which is indeed pretty stringent. Note that we also performed this detection method with a larger bin size (100 ms) in an earlier version of the manuscript and obtain similar proportion of neurons modulated. However, this larger bin size came at the expenses of a poorer temporal resolution of the laser start response which is the reason why we used a 50 ms bin size.

References:

Martiros, N., Burgess, A.A., and Graybiel, A.M. (2018). Inversely Active Striatal Projection Neurons and Interneurons Selectively Delimit Useful Behavioral Sequences. *Curr. Biol.* 28, 560-573.e5.

Mathai, A., Ma, Y., Paré, J.-F., Villalba, R.M., Wichmann, T., and Smith, Y. (2015). Reduced cortical innervation of the subthalamic nucleus in MPTP-treated parkinsonian monkeys. *Brain* 138, 946–962.

Nambu, a, Tokuno, H., Hamada, I., Kita, H., Imanishi, M., Akazawa, T., Ikeuchi, Y., and Hasegawa,

N. (2000). Excitatory cortical inputs to pallidal neurons via the subthalamic nucleus in the monkey. *J. Neurophysiol.* 84, 289–300.

4. *Line 168: typo, reproduced should be reproduce.*

Response: The typo has been corrected.

5. *Line 362: typo, compare should be compared.*

Response: The typo has been corrected.

REVIEWERS' COMMENTS:

Reviewer #1 (Remarks to the Author):

The authors have added a number of new supplementary figures to address questions and concerns of reviewers. The new figure S10 give a nice overall schematic of the hypotheses tested to provide greater context for a general audience. Clarification of the nature of the "activated anesthetized state" in parkinsonian animals has been provided. The paper adds to others from this group that have launched an important reconceptualization of the central role of the external pallidum in basal ganglia dynamics underlying the parkinsonian state. The prototypic/archypallidal framework for pallidum is extensively integrated into the experiments which is a strength. The relationship of the activated anesthetized state in rodents to the awake behaving state is not entirely clear, but the authors' very detailed optogenetic circuit deconstruction in this particular model is nevertheless an important contribution and some parallel experiments in driving oscillatory activity in normal animals helps to support generalizability of results

Response to Reviewers

Reviewer #1 Comments:

The authors have added a number of new supplementary figures to address questions and concerns of reviewers. The new figure S10 give a nice overall schematic of the hypotheses tested to provide greater context for a general audience. Clarification of the nature of the "activated anesthetized state" in parkinsonian animals has been provided. The paper adds to others from this group that have launched an important reconceptualization of the central role of the external pallidum in basal ganglia dynamics underlying the parkinsonian state. The prototypic/archypallidal framework for pallidum is extensively integrated into the experiments which is a strength. The relationship of the activated anesthetized state in rodents to the awake behaving state is not entirely clear, but the authors' very detailed optogenetic circuit deconstruction in this particular model is nevertheless an important contribution and some parallel experiments in driving oscillatory activity in normal animals helps to support generalizability of results.

Response: We thank the reviewer for the positive appraisal. We are happy to read that we have addressed all the concerns raised by all previous reviewers.